# Sparse Coding with Multi-layer Decoders using Variance Regularization

**Katrina Evtimova**                                                                                                                        *kve216@nyu.edu*
*Center for Data Science*
*New York University*

**Yann LeCun**                                                                                                                              *yann@cs.nyu.edu*
*Courant Institute and Center for Data Science*
*New York University*
*Meta AI - FAIR*

**Reviewed on OpenReview:** *https://openreview.net/forum?id=4GuIi1jJ74*

## Abstract

Sparse representations of images are useful in many computer vision applications. Sparse coding with an $l_1$ penalty and a learned linear dictionary requires regularization of the dictionary to prevent a collapse in the $l_1$ norms of the codes. Typically, this regularization entails bounding the Euclidean norms of the dictionary's elements. In this work, we propose a novel sparse coding protocol which prevents a collapse in the codes without the need to regularize the decoder. Our method regularizes the codes directly so that each latent code component has variance greater than a fixed threshold over a set of sparse representations for a given set of inputs. Furthermore, we explore ways to effectively train sparse coding systems with multi-layer decoders since they can model more complex relationships than linear dictionaries. In our experiments with MNIST and natural image patches, we show that decoders learned with our approach have interpretable features both in the linear and multi-layer case. Moreover, we show that sparse autoencoders with multi-layer decoders trained using our variance regularization method produce higher quality reconstructions with sparser representations when compared to autoencoders with linear dictionaries. Additionally, sparse representations obtained with our variance regularization approach are useful in the downstream tasks of denoising and classification in the low-data regime.

## 1 Introduction

Finding representations of data which are useful for downstream applications is at the core of machine learning and deep learning research (Bengio et al., 2013). Self-supervised learning (SSL) is a branch of representation learning which is entirely data driven and in which the representations are learned without the need for external supervision such as semantic annotations. In SSL, learning can happen through various pre-text tasks such as reconstructing the input with autoencoders (Vincent et al., 2008), predicting the correct order of frames in a video (Misra et al., 2016; Fernando et al., 2017), enforcing equivariant representations across transformations of the same image (He et al., 2019; Grill et al., 2020; Zbontar et al., 2021; Bardes et al., 2021) or enforcing other invariance properties (Földiák, 1991; Wiskott & Sejnowski, 2002; Hyvärinen et al., 2003; Chen et al., 2018). The pre-text tasks should be designed to extract data representations useful in other downstream tasks.

We focus on learning representations of images through *sparse coding*, a self-supervised learning paradigm based on input reconstruction (Olshausen & Field, 1996). A representation (or *code*) in the form of a feature vector $z \in \mathbb{R}^l$ is considered sparse if only a small number of its components are active for a given data sample. Sparsity introduces the inductive bias that only a small number of factors are relevant for a single data point

and it is a desirable property for data representations as it can improve efficiency and interpretability of the representations (Bengio, 2009; Bengio et al., 2013). While it is an open debate, sparse representations of images have been shown to improve classification performance (Ranzato et al., 2007; Kavukcuoglu et al., 2010; Makhzani & Frey, 2013) and are popular for denoising and inpainting applications (Elad & Aharon, 2006; Mairal et al., 2007). They are biologically motivated as the primary visual cortex uses sparse activations of neurons to represent natural scenes (Olshausen & Field, 1996; Yoshida & Ohki, 2020).

In sparse coding, the goal is to find a latent sparse representation $z \in \mathbb{R}^l$ which can reconstruct an input $y \in \mathbb{R}^d$ using a *learned decoder* $\mathcal{D}$. Typically, the decoder is linear, namely $\mathcal{D} \in \mathbb{R}^{d \times l}$, and the sparse code $z$ selects a linear combination of a few of its columns to reconstruct the input. In practice, a sparse code $z$ is computed using an *inference algorithm* which minimizes $z$'s $l_1$ norm and simultaneously optimizes for a good reconstruction of the input $y$ using $\mathcal{D}$'s columns. Learning of the decoder $\mathcal{D}$ typically happens in an iterative EM-like fashion. Inference provides a set of sparse codes corresponding to some inputs (expectation step). Given these codes, the decoder's parameters are updated using gradient descent to minimize the error between the original inputs and their reconstructions (maximization step). Sparse coding systems in which the decoder is non-linear are hard to learn and often require layer-wise training (Zeiler et al., 2010; He et al., 2014). In this work, we explore ways to effectively train sparse coding systems with non-linear decoders using end-to-end learning.

In order to successfully train a sparse coding system which learns a linear dictionary $\mathcal{D}$, it is common to bound the norms of the dictionary's weights in order to avoid collapse in the $l_1$ norms of the codes. Indeed, consider a reconstruction $\tilde{y}$ obtained from a dictionary $\mathcal{D}$ and code $z$, namely $\tilde{y} = \mathcal{D}z$. The same reconstruction can be obtained from a re-scaled dictionary $\mathcal{D}' = c\mathcal{D}$ and code $z' = z/c$ for any non-zero multiple $c$. When $c > 1$, the representation $z'$ has a smaller $l_1$ norm than $z$ but produces the same reconstruction. In practice, if $\mathcal{D}$'s weights are unbounded during training, they become arbitrarily large, while the $l_1$ norms of the latent representations become arbitrarily small which leads to a collapse in their $l_1$ norm. Typically, in order to restrict a linear dictionary $\mathcal{D}$, its columns are re-scaled to have a constant $l_2$ norm (Lee et al., 2006) or weight decay is applied to its weights (Sun et al., 2018). However, in the case when $\mathcal{D}$ is a multi-layer neural network trained end-to-end, it is not clear what the best strategy to bound the decoder's weights is in order to avoid collapse in the codes and at the same time learn useful features.

Instead of restricting $\mathcal{D}$'s weights, we propose to prevent collapse in the $l_1$ norm of the codes directly by applying variance regularization to each latent code component. In particular, we add a regularization term to the energy minimized during inference for computing a set of codes. This term encourages the variance of each latent code component to be greater than a fixed lower bound. This strategy is similar to the variance principle presented in Bardes et al. (2021) and related to methods that directly regularize the latent codes (Földiak, 1990; Olshausen & Field, 1996; Barello et al., 2018). Our strategy ensures that the norms of the sparse codes do not collapse and we find that it removes the need to explicitly restrict $\mathcal{D}$'s parameters. Since this variance regularization term is independent from $\mathcal{D}$'s architecture, we can use it when the decoder is a deep neural network. In this work, we experiment with fully connected sparse autoencoders with a one hidden layer decoder and successfully train such models using our variance regularization strategy.

The main contributions of our work are:

- We introduce an effective way to train sparse autoencoders and prevent collapse in the $l_1$ norms of the codes that they produce without the need to regularize the decoder's weights but instead by encouraging the latent code components to have variance above a fixed threshold.

- We show empirically that linear decoders trained with this variance regularization strategy are comparable to the ones trained using the standard $l_2$-normalization approach in terms of the features that they learn and the quality of the reconstructions they produce. Moreover, variance regularization successfully extends sparse coding to the case of a non-linear fully connected decoder with one hidden layer.

- Additionally, our experiments show that sparse representations obtained from autoencoders with a non-linear decoder trained using our proposed variance regularization strategy: 1) produce higher quality reconstructions than ones obtained from autoencoders with a linear decoder, 2) are helpful in the down-stream task of classification in the low-data regime.

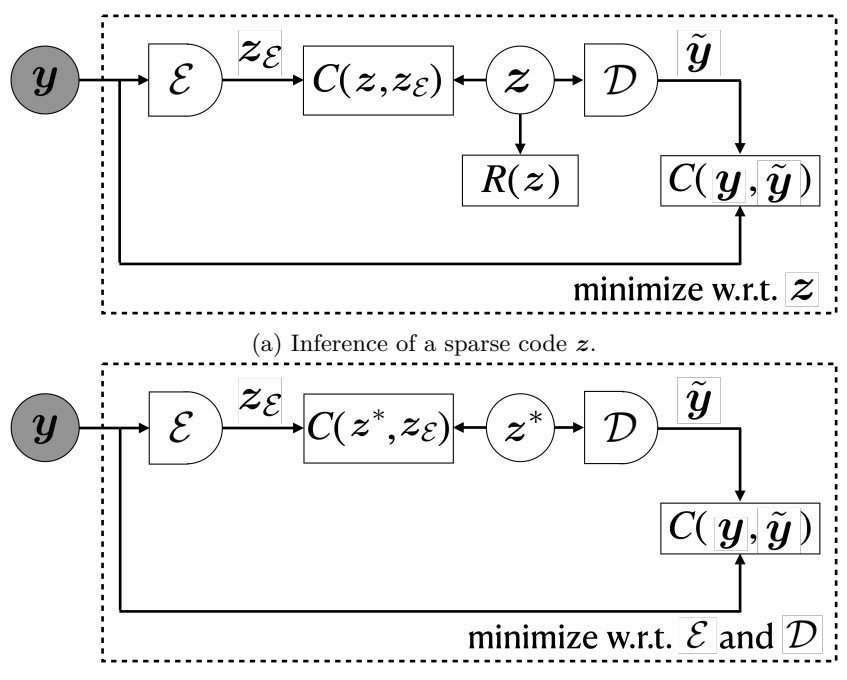

(a) Inference of a sparse code $z$.

(b) Learning of the encoder $\mathcal{E}$ and decoder $\mathcal{D}$.

Figure 1: Our sparse autoencoder setup. $C$ denotes mean squared error. The inference step in Figure 1a finds a code $z^*$ which minimizes the reconstruction error between the input $y$ and its reconstruction $\tilde{y}$. The code is subject to sparsity and variance regularization, denoted by $R(z)$. Additionally, the code is regularized to stay close to the encoder's predictions $z_{\mathcal{E}}$. During the learning step in Figure 1b, the encoder $\mathcal{E}$ is trained to predict the codes $z^*$ computed during inference and the decoder $\mathcal{D}$ learns to reconstruct inputs $y$ from the sparse codes $z^*$.

## 2  Method

A schematic view of our sparse autoencoder setup is presented in Figure 1. At a high level, given an input $y$ and a fixed decoder $\mathcal{D}$, we perform inference using a modified version of the FISTA algorithm (Beck & Teboulle, 2009) to find a sparse code $z^*$ which can best reconstruct $y$ using $\mathcal{D}$'s elements. The decoder's weights are trained by minimizing the mean squared error between the input $y$ and the reconstruction $\tilde{y}$ computed from $z^*$. The encoder $\mathcal{E}$ is trained to predict the codes $z^*$ computed during inference. More details on each of the components in Figure 1 are presented in the following sections.

### 2.1  Background: Sparse Coding and Dictionary Learning

**Inference**    Sparse coding algorithms with an $l_1$ sparsity penalty and a linear decoder $\mathcal{D} \in \mathbb{R}^{d \times l}$ perform inference to find a latent sparse representation $z \in \mathbb{R}^l$ of a given data sample $y \in \mathbb{R}^d$ which minimizes the following energy function:

$$\mathbf{E}(z, y, \mathcal{D}) = \frac{1}{2}\|y - \mathcal{D}z\|_2^2 + \lambda\|z\|_1. \tag{1}$$

The first term in (1) is the reconstruction error for the sample $y$ using code $z$ and decoder $\mathcal{D}$. The second term is a regularization term which penalizes the $l_1$ norm of $z$. In essence, each code $z$ selects a linear combination of the decoder's columns. The decoder can thus be thought of as a *dictionary* of building blocks. Inference algorithms find a solution for the minimization problem in (1) keeping $\mathcal{D}$ fixed, namely

$$z^* = \arg\min_{z} \mathbf{E}(z, y, \mathcal{D}).$$

**Learning the Decoder** The decoder's parameters can be learned through gradient-based optimization by minimizing the mean squared reconstruction error between elements in the training set $\mathcal{Y} = \{\boldsymbol{y}_1, \ldots, \boldsymbol{y}_N\} \subset \mathbb{R}^d$ and their corresponding reconstructions from the sparse representations $\mathcal{Z}^* = \{\boldsymbol{z}_1^*, \ldots, \boldsymbol{z}_N^*\} \subset \mathbb{R}^l$ obtained during inference, namely:

$$\arg\min_{\mathcal{D}} \mathcal{L}_{\mathcal{D}}(\mathcal{D}, \mathcal{Z}^*, \mathcal{Y}) = \arg\min_{\mathcal{D}} \frac{1}{N} \sum_{i=1}^{N} \|\boldsymbol{y}_i - \mathcal{D}\boldsymbol{z}_i^*\|_2^2, \tag{2}$$

where $\mathcal{D}$'s columns are typically projected onto the unit sphere to avoid collapse in the latent codes, namely $\|\mathcal{D}_{:,i}\|_2^2 = 1$ for $\mathcal{D}_{:,i} \in \mathbb{R}^d$.

## 2.2 Background: Inference with ISTA and FISTA

We provide an overview of the Iterative Shrinkage-Thresholding Algorithm (ISTA) and its faster version FISTA which we adopt in our setup. ISTA is a proximal gradient method which performs inference to find a sparse code $\boldsymbol{z}^*$ which minimizes the energy in (1) for a given input $\boldsymbol{y}$ and a fixed dictionary $\mathcal{D}$. We can re-write the inference minimization problem as:

$$\boldsymbol{z}^* = \arg\min_{\boldsymbol{z}} \mathbf{E}(\boldsymbol{z}, \boldsymbol{y}, \mathcal{D}) = \arg\min_{\boldsymbol{z}} f(\boldsymbol{z}) + g(\boldsymbol{z}), \tag{3}$$

where $f(\boldsymbol{z}) := \frac{1}{2}\|\boldsymbol{y} - \mathcal{D}\boldsymbol{z}\|_2^2$ is the reconstruction term and $g(\boldsymbol{z}) := \lambda\|\boldsymbol{z}\|_1$ is the $l_1$ penalty term. In ISTA, the code $\boldsymbol{z}$ can be given any initial value and is typically set to the zero vector: $\boldsymbol{z}^{(0)} = \boldsymbol{0} \in \mathbb{R}^l$. An ISTA iteration consists of two steps: a gradient step and a shrinkage step.

**Gradient Step** To compute code $\boldsymbol{z}^{(k)}$ from its previous iteration $\boldsymbol{z}^{(k-1)}$, where $k \in \{1, \ldots, K\}$, ISTA first performs the gradient update by taking gradient step of size $\eta_k$ with respect to the squared error between the input and the reconstruction from $\boldsymbol{z}^{(k-1)}$:

$$\text{ISTA gradient step: } \tilde{\boldsymbol{z}}^{(k)} = \boldsymbol{z}^{(k-1)} - \eta_k \nabla f(\boldsymbol{z}^{(k-1)}), \tag{4}$$

The size of the gradient step $\eta_k$ at iteration $k$ can be determined with backtracking (Beck & Teboulle, 2009).

**Shrinkage Step** The shrinkage step is applied to the intermediate output from the gradient step in ISTA. It uses the shrinkage function $\tau_\alpha$ which sets components of its input to 0 if their absolute value is less than $\alpha$ or otherwise contracts them by $\alpha$, namely $[\tau_\alpha(\boldsymbol{x})]_j = \text{sign}(x_j)(|x_j| - \alpha)_+$ for some threshold $\alpha > 0$ where $j$ traverses the components of its input vector $\boldsymbol{x}$. The shrinkage step for ISTA is given by:

$$\text{ISTA shrinkage step: } \boldsymbol{z}^{(k)} = \tau_{\lambda\eta_k}(\tilde{\boldsymbol{z}}^{(k)}) \tag{5}$$

where $\lambda > 0$ is the hyperparameter which determines the sparsity level. In the case when the codes are restricted to be non-negative, a property that we adopt in our setup, the shrinkage function can be modified to set all negative components in its output to 0: $[\tilde{\tau}_\alpha(\boldsymbol{x})]_j = ([\tau_\alpha(\boldsymbol{x})]_j)_+$.

**FISTA** In our setup, we use FISTA (Beck & Teboulle, 2009) which accelerates the convergence of ISTA by adding a momentum term in each of its iterations. Namely, $\boldsymbol{z}^{(k-1)}$ in (4) is replaced by:

$$\boldsymbol{x}^{(k)} = \boldsymbol{z}^{(k-1)} + \frac{t_{k-1} - 1}{t_k}(\boldsymbol{z}^{(k-1)} - \boldsymbol{z}^{(k-2)}) \tag{6}$$

for $k \geq 2$ where $t_k = \frac{1 + \sqrt{1 + 4t_{k-1}^2}}{2}$ and $t_1 = 1$. Thus, the FISTA gradient step becomes:

$$\tilde{\boldsymbol{z}}^{(k)} = \boldsymbol{x}^{(k)} - \eta_k \nabla f(\boldsymbol{x}^{(k)}). \tag{7}$$

FISTA employs the same shrinkage step as ISTA.

### 2.3 Modified Inference with Variance Regularization on the Latent Code Components

In order to prevent collapse in the $l_1$ norm of the latent codes, we propose to ensure that the variance of each latent code component stays above a pre-set threshold. To achieve this, we add a regularization term to the energy in (3) which encourages the variance of all latent component across a mini-batch of codes to be greater than a pre-set threshold. In particular, we replace the function $f(\boldsymbol{z}) = \frac{1}{2}\|\boldsymbol{y} - \mathcal{D}\boldsymbol{z}\|_2^2$ in (3) with a new function $\tilde{f}(\boldsymbol{Z})$ defined over a set of codes $\boldsymbol{Z} \in \mathbb{R}^{l \times n}$ corresponding to a mini-batch of data samples $Y = \{\boldsymbol{y}_1, \ldots, \boldsymbol{y}_n\}$ as:

$$\tilde{f}(\boldsymbol{Z}) = \sum_{i=1}^{n} \frac{1}{2}\|\boldsymbol{y}_i - \mathcal{D}\boldsymbol{Z}_{:,i}\|_2^2 + \sum_{j=1}^{l} \beta\left[\left(T - \sqrt{\mathrm{Var}(\boldsymbol{Z}_{j,:})}\right)_+\right]^2. \tag{8}$$

The first sum in (8) is over the reconstruction terms from each code $\boldsymbol{Z}_{:,i} \in \mathbb{R}^l$. The second sum in (8) is over squared hinge terms involving the variance of each latent component $\boldsymbol{Z}_{j,:} \in \mathbb{R}^n$ across the batch where $\mathrm{Var}(\boldsymbol{Z}_{j,:}) = \frac{1}{n-1}\sum_{i=1}^{n}(\boldsymbol{Z}_{j,i} - \mu_j)^2$ and $\mu_j$ is the mean across the $j$-th latent component, namely $\mu_j = \frac{1}{n}\sum_{i=1}^{n} Z_{j,i}$. The hinge terms are non-zero for any latent dimension whose variance is below the fixed threshold of $\sqrt{T}$.

**Modified Energy: Variance Regularization**  Using $\tilde{f}(\boldsymbol{Z})$ from (8) and setting $\tilde{g}(\boldsymbol{Z}) := \sum_{i=1}^{n} g(\boldsymbol{Z}_{:,i})$, our modified objective during inference is to minimize the energy:

$$\tilde{\mathbf{E}}(\boldsymbol{Z}, Y, \mathcal{D}) = \tilde{f}(\boldsymbol{Z}) + \tilde{g}(\boldsymbol{Z}) = \sum_{i=1}^{n} \frac{1}{2}\|\boldsymbol{y}_i - \mathcal{D}\boldsymbol{Z}_{:,i}\|_2^2 + \sum_{j=1}^{l} \beta\left[\left(T - \sqrt{\mathrm{Var}(\boldsymbol{Z}_{j,:})}\right)_+\right]^2 + \sum_{i=1}^{n} \lambda\|\boldsymbol{Z}_{:,i}\|_1 \tag{9}$$

with respect to the codes $\boldsymbol{Z}$ for a mini-batch of samples $Y$ and a fixed dictionary $\mathcal{D}$. Note that the hinge terms counteract with the $l_1$ penalty terms in (9). The hinge terms act as regularization terms which encourage the variance of each of the latent code components to remain above the threshold of $\sqrt{T}$. This should prevent a collapse in the $l_1$ norm of the latent codes directly and thus remove the need to normalize the weights of the decoder.

**Gradient Step**  The first step in our modified version of FISTA is to take a gradient step for each code $\boldsymbol{Z}_{:,t} \in \mathbb{R}^l$:

$$\tilde{\boldsymbol{Z}}_{:,t}^{(k)} = \boldsymbol{Z}_{:,t}^{(k-1)} - \eta_k \nabla_{\boldsymbol{Z}_{:,t}^{(k-1)}} \tilde{f}(\boldsymbol{Z}). \tag{10}$$

In practice, since we use FISTA, the code $\boldsymbol{Z}_{:,t}^{(k-1)}$ in (10) is replaced by a momentum term as defined in (6) but here we omit this notation for simplicity. The gradient of the sum of reconstruction terms in (8) with respect to $Z_{s,t}$, one of the latent components of code $\boldsymbol{Z}_{:,t}$, for a linear decoder $\mathcal{D}$ is:

$$\frac{\partial}{\partial Z_{s,t}} \sum_{i=1}^{n} \frac{1}{2}\|\boldsymbol{y}_i - \mathcal{D}\boldsymbol{Z}_{:,i}\|_2^2 = \mathcal{D}_s^T(\mathcal{D}\boldsymbol{Z}_{:,t} - \boldsymbol{y}_t), \tag{11}$$

where $\mathcal{D}_s$ denotes the $s$-th column of the dictionary $\mathcal{D}$.[1] The gradient of the sum of hinge terms in (8) with respect to $Z_{s,t}$ is:

$$\frac{\partial}{\partial Z_{s,t}} \sum_{j=1}^{l} \beta\left[\left(T - \sqrt{\mathrm{Var}(\boldsymbol{Z}_{j,:})}\right)_+\right]^2 = \begin{cases} -\frac{2\beta}{n-1}\frac{T - \sqrt{\mathrm{Var}(\boldsymbol{Z}_{s,:})}}{\sqrt{\mathrm{Var}(\boldsymbol{Z}_{s,:})}}(Z_{s,t} - \mu_s), & \text{if } \sqrt{\mathrm{Var}(\boldsymbol{Z}_{s,:})} < T \\ 0 & \text{otherwise.} \end{cases} \tag{12}$$

---

[1]The gradient for any neural network decoder $\mathcal{D}$ can be computed through automatic differentiation.

Even though the hinge terms in (8) are not smooth convex functions[2], the fact that the gradient in (12) is a line implies that the hinge term behaves locally as a convex quadratic function. A visualization of $h(\boldsymbol{x}) = \left[\left(1 - \sqrt{\mathrm{Var}(\boldsymbol{x})}\right)_+\right]^2$ for $\boldsymbol{x} \in \mathbb{R}^2$ and the full derivation of (12) are available in Appendix A.2 and A.3, respectively.

**Shrinkage Step**    Similarly to the regular FISTA protocol, the shrinkage step in our modified FISTA is applied to the intermediate results from the gradient step in (10).

### 2.4    Encoder for Amortized Inference

One limitation of the proposed variance regularization is that the modified inference procedure depends on batch statistics (see equation 12) and thus the codes are not deterministic. To address this limitation, we propose to train an encoder $\mathcal{E}$ simultaneously with the decoder $\mathcal{D}$ to predict the sparse codes computed from inference with our modified FISTA. After training of the encoder and decoder is done, the encoder can be used to compute sparse codes for different inputs independently from each other in a deterministic way. Additionally, using an encoder reduces the inference time: instead of performing inference with FISTA, the encoder computes sparse representations directly from the inputs and thus provides *amortized inference*.

**Learning the Encoder**    The encoder is trained using mini-batch gradient descent to minimize the following mean-squared error between its predictions and the sparse codes $\boldsymbol{Z}^* \in \mathbb{R}^{l \times n}$ computed from inference with FISTA for inputs $Y = \{\boldsymbol{y}_1, \ldots, \boldsymbol{y}_n\}$:

$$\arg\min_{\mathcal{E}} \mathcal{L}_{\mathcal{E}}(\mathcal{E}, \boldsymbol{Z}^*, Y) = \arg\min_{\mathcal{E}} \frac{1}{n} \sum_{i=1}^{n} \|\mathcal{E}(\boldsymbol{y}_i) - \boldsymbol{Z}^*_{:,i}\|_2^2. \tag{13}$$

Note that the codes $\boldsymbol{Z}^*$ are treated as constants. Training the encoder this way is similar to target propagation (Lee et al., 2015).

**Modified Energy: Variance and Encoder Regularization**    To encourage FISTA to find codes which can be learned by the encoder, we further modify the inference protocol by adding a term to $\tilde{f}(\boldsymbol{Z})$ which penalizes the distance between FISTA's outputs and the encoder's outputs:

$$\tilde{f}(\boldsymbol{Z}) = \sum_{i=1}^{n} \frac{1}{2}\|\boldsymbol{y}_i - \mathcal{D}\boldsymbol{Z}_{:,i}\|_2^2 + \beta \sum_{j=1}^{l} \left[\left(T - \sqrt{\mathrm{Var}(\boldsymbol{Z}_{j,:})}\right)_+\right]^2 + \gamma \sum_{i=1}^{n} \|\boldsymbol{Z}_{:,i} - \mathcal{E}(\boldsymbol{y}_i)\|_2^2. \tag{14}$$

This regularization term ensures that the codes computed during inference with FISTA do not deviate too much from the encoder's predictions. In our setup, the encoder's predictions are treated as constants and are used as the initial value for codes in FISTA, namely $\boldsymbol{Z}^{(0)}_{:,i} := \mathcal{E}(\boldsymbol{y}_i)$. This can reduce the inference time by lowering the number of FISTA iterations needed for convergence if the encoder provides good initial values. With the new $\tilde{f}(\boldsymbol{Z})$ from equation 14, the energy minimized during inference becomes:

$$\tilde{\mathbf{E}}(\boldsymbol{Z}, Y, \mathcal{D}) = \sum_{i=1}^{n} \frac{1}{2}\|\boldsymbol{y}_i - \mathcal{D}\boldsymbol{Z}_{:,i}\|_2^2 + \beta \sum_{j=1}^{l} \left[\left(T - \sqrt{\mathrm{Var}(\boldsymbol{Z}_{j,:})}\right)_+\right]^2 + \gamma \sum_{i=1}^{n} \|\boldsymbol{Z}_{:,i} - \mathcal{E}(\boldsymbol{y}_i)\|_2^2 + \sum_{i=1}^{n} \lambda\|\boldsymbol{Z}_{:,i}\|_1.$$
$$\tag{15}$$

---

[2]A requirement for the FISTA algorithm.

## 3 Experimental Setup

### 3.1 Encoder and Decoder Architectures

**LISTA Encoder**   The encoder's architecture is inspired by the Learned ISTA (LISTA) architecture (Gregor & LeCun, 2010). LISTA is designed to mimic outputs from inference with ISTA and resembles a recurrent neural network. Our encoder consists of two fully connected layers $U \in \mathbb{R}^{d \times l}$ and $S \in \mathbb{R}^{l \times l}$, a bias term $b \in \mathbb{R}^{l}$, and ReLU activation functions. Algorithm 1 describes how the encoder's outputs are computed. Our version of LISTA produces non-negative codes by design. We refer to our encoder as LISTA in the rest of the paper.

---

**Algorithm 1** LISTA encoder $\mathcal{E}$: forward pass

**Input:** Image $y \in \mathbb{R}^d$, number of iterations $L$
**Parameters:** $U \in \mathbb{R}^{d \times l}$, $S \in \mathbb{R}^{l \times l}$, $b \in \mathbb{R}^l$
**Output:** sparse code $z_{\mathcal{E}} \in \mathbb{R}^l$
$u = Uy + b$
$z_0 = \mathrm{ReLU}(u)$
**for** $i = 1$ **to** $L$ **do**
    $z_i = \mathrm{ReLU}(u + Sz_{i-1})$
**end for**
$z_{\mathcal{E}} = z_L$

---

**Linear Decoder**   A linear decoder $\mathcal{D}$ is parameterized simply as a linear transformation $W$ which maps codes in $\mathbb{R}^l$ to reconstructions in the input data dimension $\mathbb{R}^d$, i.e. $W \in \mathbb{R}^{d \times l}$. No bias term is used in this linear transformation.

**Non-Linear Decoder**   In the case of a non-linear decoder $\mathcal{D}$, we use a fully connected network with one hidden layer of size $m$ and ReLU as an activation function. We refer to the layer which maps the input codes to hidden representations as $W_1 \in \mathbb{R}^{m \times l}$ and the one which maps the hidden representations to the output (reconstructions) as $W_2 \in \mathbb{R}^{d \times m}$. There is a bias term $b_1 \in \mathbb{R}^m$ following $W_1$ and no bias term after $W_2$. Given an input $z \in \mathbb{R}^l$, the decoder's output is computed as: $\tilde{y} = W_2(W_1z + b_1)_+$.

### 3.2 Evaluation

We compare sparse autoencoders trained using our variance regularization method to sparse autoencoders trained using other regularization methods. We refer to our approach as *variance-regularized dictionary learning* (**VDL**) in the case when the decoder is a linear dictionary and as **VDL-NL** when the decoder is non-linear. We refer to the traditional approach in which the $l_2$ norm of the decoder's columns is fixed as *standard dictionary learning* (**SDL**) in the case the decoder is linear and as **SDL-NL** in the case it is non-linear - a natural extension of SDL to the case of non-linear decoders is to fix the $l_2$ norms of the columns of both layers $W_1$ and $W_2$. Additionally, we compare our variance regularization method to sparse autoencoders in which only weight decay is used to avoid collapse which we refer to as **WDL** and **WDL-NL** in the case of a linear and a non-linear decoder, respectively. Finally, we consider a *decoder-only* setup in which the standard approach of fixing the $l_2$ norm of the decoder's columns is applied and the codes are computed using inference with FISTA. We refer to this setup as **DO** when the decoder is linear and as **DO-NL** when the decoder is non-linear.

In order to compare the different models, we train them with different levels of sparsity regularization $\lambda$, evaluate the quality of their reconstructions in terms of peak signal-to-noise ratio (PSNR), and visualize the features that they learn. We evaluate models with a linear decoder on the downstream task of denoising. Additionally, we measure the pre-trained models' performance on the downstream task of MNIST classification in the low data regime.

### 3.3 Data Processing

**MNIST**   In the first set of our experiments, we use the MNIST dataset (LeCun & Cortes, 2010) consisting of $28 \times 28$ hand-written digits and do standard pre-processing by subtracting the global mean and dividing by the global standard deviation. We split the training data randomly into 55000 training samples and 5000 validation samples. The test set consists of 10000 images.

**Natural Image Patches**   For experiments with natural images, we use patches from ImageNet ILSVRC-2012 (Deng et al., 2009). We first convert the ImageNet images to grayscale and then do standard pre-

processing by subtracting the global mean and dividing by the global standard deviation. We then apply local contrast normalization to the images with a $13 \times 13$ Gaussian filter with standard deviation of 5 following Jarrett et al. (2009) and Zeiler et al. (2011). We use 200000 randomly selected patches of size $28 \times 28$ for training, 20000 patches for validation, and 20000 patches for testing.

### 3.4 Inference and Training

We determine all hyperparameter values through grid search. Full training details can be found in Appendix B.1.

**Inference** The codes $\boldsymbol{Z}$ are restricted to be non-negative as described in section 2.2. The dimension of the latent codes in our MNIST experiments is $l = 128$ and in experiments with ImageNet patches it is $l = 256$. The batch size is set to 250 which we find sufficiently large for the regularization term on the variance of each latent component in (8) for VDL and VDL-NL models. We set the maximum number of FISTA iterations $K$ to 200 which we find sufficient for good reconstructions.

**Autoencoders Training** We train models for a maximum of 200 epochs in MNIST experiments and for 100 epochs in experiments with natural image patches. We apply early stopping and save the autoencoder which outputs codes with the lowest average energy measured on the validation set.[3] In SDL and SDL-NL experiments, we fix the $l_2$ norms of the columns in the decoders' fully connected layers $\boldsymbol{W}$, $\boldsymbol{W}_1$, and $\boldsymbol{W}_2$ to be equal to 1. We add weight decay to the bias term $\boldsymbol{b}_1$ in models with non-linear decoder as well as to the bias term $\boldsymbol{b}$ in LISTA encoders to prevent their norms from inflating arbitrarily.

**Implementation and Hardware** Our PyTorch implementation is available on GitHub at https://github.com/kevtimova/deep-sparse. We train our models on one NVIDIA RTX 8000 GPU card and all our experiments take less than 24 hours to run.

## 4 Results and Analysis

### 4.1 Visualizing Learned Features

**Linear Decoders** Figures 2 shows the dictionary elements for two SDL and two VDL decoders trained with different levels of sparsity regularization $\lambda$. As evident from Figures 2a and 2b we confirm that, in the case of standard dictionary learning on MNIST, the dictionary atoms look like orientations, strokes, and parts of digits for lower values of the sparsity regularization $\lambda$ and they become looking like full digit prototypes as $\lambda$ increases (Yu, 2012). A similar transition from strokes to full digit prototypes is observed in figures 2c and 2d displaying the dictionary elements of models trained using variance regularization in the latent code components. The same trend is observed in dictionary elements from WDL and DO models (Figure 12 in the Appendix). Figure 3 shows dictionary elements for two SDL and two VDL models trained on ImageNet patches. All of these models contain gratings and Gabor-like filters. We observe that models which are trained with a lower sparsity regularization parameter $\lambda$ (Figures 3a and 3c) contain more high frequency atoms than ones trained with higher values of $\lambda$ (Figures 3b and 3d).

**Non-linear Decoders** Each column $\boldsymbol{W}_{:,i} \in \mathbb{R}^d$ in a linear dictionary $\mathcal{D}$ (parameterized by $\boldsymbol{W}$) reflects what a single latent component encodes since $\boldsymbol{W}_{:,i} = W\boldsymbol{e}^{(i)}$ where $\boldsymbol{e}^{(i)} \in \mathbb{R}^l$ is a one-hot vector with 1 in position $i$ and 0s elsewhere. We use a similar strategy to visualize the features that non-linear decoders learn. Namely, we provide codes $\boldsymbol{e}^{(i)}$ with only one non-zero component as inputs to the non-linear decoders and display the resulting "atoms" in image space.[4] While in the linear case each code component selects a single dictionary column, in the case of non-linear decoders with one hidden layer, each code component selects a linear combination of columns in $\boldsymbol{W}_2$. Two SDL-NL and two VDL-NL models trained on MNIST with different levels of sparsity regularization $\lambda$ yield the atoms displayed in Figure 4. They contain strokes and orientations for smaller values of $\lambda$ and fuller digit prototypes for larger values of $\lambda$ and are very similar to the dictionary elements from linear decoders in Figure 2. We use the same strategy to visualize the features that non-linear decoders trained on ImageNet patches learn. These features are displayed in Figure 5 for

---

[3]As a reminder, codes are computed using amortized inference with the encoder during validation.

[4]In practice, we remove the effect of the bias term $b_1$ after $W_1$ and display $\mathcal{D}(\boldsymbol{e}^{(i)}) - \mathcal{D}(\boldsymbol{0})$.

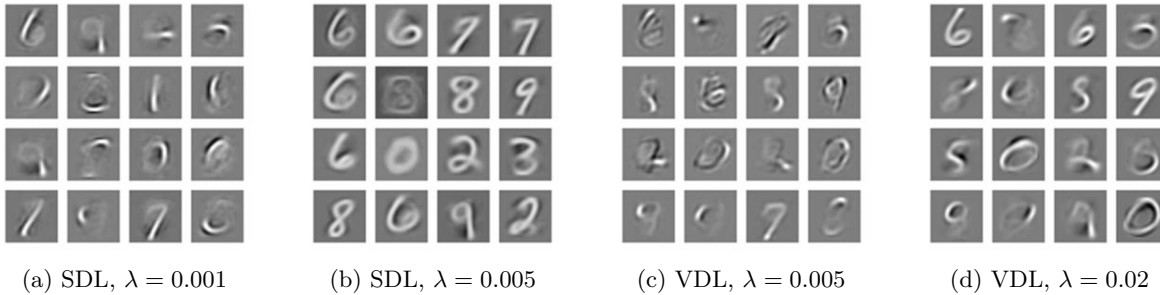

| (a) SDL, $\lambda = 0.001$ | (b) SDL, $\lambda = 0.005$ | (c) VDL, $\lambda = 0.005$ | (d) VDL, $\lambda = 0.02$ |

Figure 2: Dictionary elements for linear dictionaries from SDL and VDL models with latent dimension $l = 128$ and different levels of sparsity regularization $\lambda$. The SDL models in (2a) and (2b) produce reconstructions with PSNR of 21.1 and 18.6 for codes with average sparsity level of 63% and 83% on the test set, respectively. The VDL models in (2c) and (2d) produce reconstructions with PSNR of 20.7 and 17.7 for codes with average sparsity level of 69% and 91.8% on the test set, respectively.

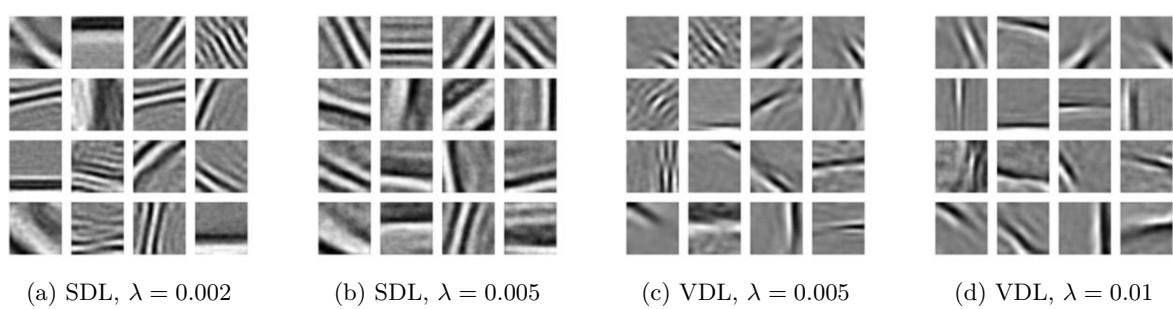

| (a) SDL, $\lambda = 0.002$ | (b) SDL, $\lambda = 0.005$ | (c) VDL, $\lambda = 0.005$ | (d) VDL, $\lambda = 0.01$ |

Figure 3: Dictionary elements which resemble gratings and Gabor filters for SDL and VDL models with latent dimension $l = 256$ trained on ImageNet patches with different levels of sparsity regularization $\lambda$. The SDL models in 3a and 3b produce reconstructions with average PSNR of 26.9 and 24.0 and average sparsity level in the codes of 74.6% and 90.5% on the test set, respectively. The VDL models in Figures 3c and 3d produce reconstructions with average PSNR of 27.3 and 25.3 and average sparsity level in the codes of 77.3% and 89.2% on the test set, respectively.

SDL-NL and VDL-NL models trained with different levels of sparsity regularization $\lambda$. As in the case of linear decoders, we observe that both SDL-NL and VDL-NL models learn gratings and Gabor-like filters.

**Encoders** Figure 14 in the Appendix shows visualizations of the $U \in \mathbb{R}^{d \times l}$ layer of the LISTA encoder in SDL, VDL, and WDL models with code dimension $l = 128$ trained on MNIST images and different levels of sparsity regularization $\lambda$. Similarly to the decoders in 11, the features learned by the encoders resemble orientations and parts of digits but in the case of WDL there is a larger proportion of noisy features compared to the other models as $\lambda$ increases.

### 4.2 Reconstruction Quality

Figure 6 plots regret curves for the trade-off between average sparsity in the codes and reconstruction quality from these codes for models with linear decoders (dashed lines) and non-linear decoders (solid lines) trained on MNIST images (Figure 6a) and ImageNet patches (Figure 6b). The sparsity level is measured by the average percentage of non-active (zero) code components and the reconstruction quality is measured by the average PSNR. Both are evaluated on the test set over 5 random seeds. As expected, higher sparsity levels result in worse reconstructions. [5] Moreover, models trained with our variance regularization approach and a non-linear decoder (VDL-NL) on both MNIST and ImageNet patches produce better reconstructions than

---

[5]As a reference for reconstruction quality in terms of PSNR, the second row of images in Figure 7a and 7b contains reconstructions of the images in the top row with PSNR of around 17.3 and 17.7, respectively.

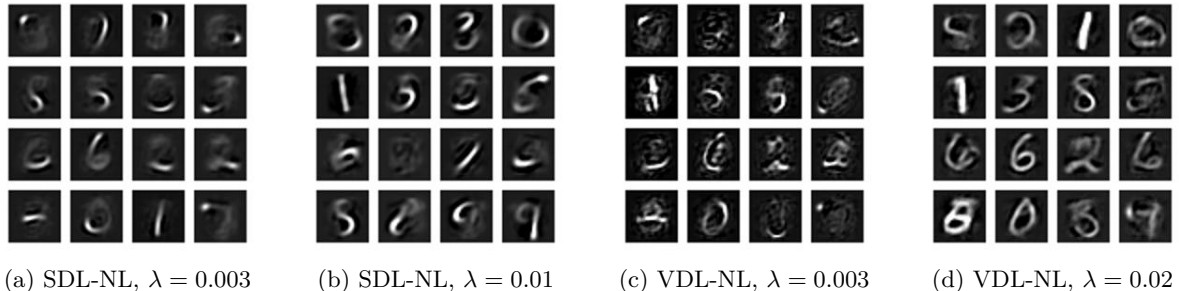

(a) SDL-NL, $\lambda = 0.003$     (b) SDL-NL, $\lambda = 0.01$     (c) VDL-NL, $\lambda = 0.003$     (d) VDL-NL, $\lambda = 0.02$

Figure 4: Visualizing reconstructions from SDL-NL and VDL-NL decoders trained on MNIST and input codes in which only one component is active and the rest are zero. The models are trained with different levels of sparsity regularization $\lambda$. Each code component encodes stokes, orientations, parts of digits, and full digit prototypes very similar to those in Figure 2. The SDL-NL models in 4a and 4b produce reconstructions with average PSNR of 20.9 and 18.4 and average sparsity level in the codes of 69.2% and 85.6% on the test set, respectively. The VDL-NL models in 4c and 4d produce reconstructions with average PSNR of 22.7 and 18.3 and average sparsity level in the codes of 58.8% and 92.2% on the test set, respectively.

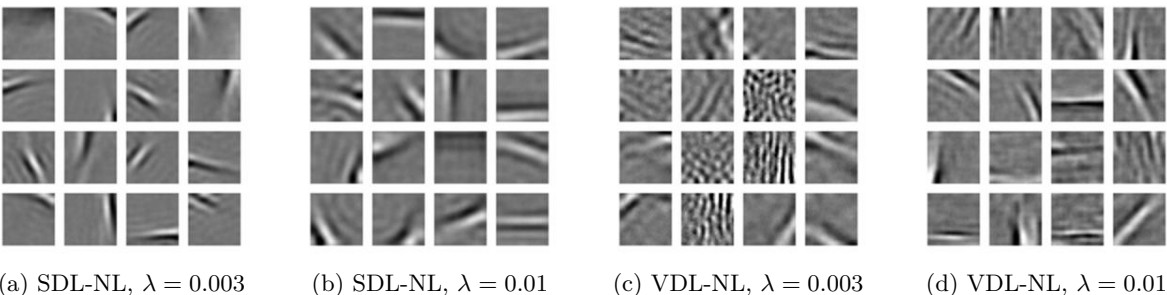

(a) SDL-NL, $\lambda = 0.003$     (b) SDL-NL, $\lambda = 0.01$     (c) VDL-NL, $\lambda = 0.003$     (d) VDL-NL, $\lambda = 0.01$

Figure 5: Visualizing reconstructions from SDL-NL and VDL-NL decoders trained on ImageNet patches and input codes in which only one code component is active and the rest are zero. The models are trained with different levels of sparsity regularization $\lambda$. Code components encode gratings and Gabor-like filters. The SDL-NL models in (5a) and (5b) produce reconstructions with PSNR of 27.1 and 25.0 for codes with average sparsity level of 69.8% and 83.4%, respectively. The VDL-NL models in (5c) and (5d) produce reconstructions with PSNR of 28.4 and 24.9 for codes with average sparsity level of 67.6% and 90.8%, respectively.

other models for higher levels of sparsity. This implies that VDL-NL models can better utilize the additional representational capacity from the larger number of layers and parameters in the non-linear decoder compared to the other models when the sparsity level is high.

## 4.3 Denoising

We evaluate the performance of SDL and VDL autoencoders on the downstream task of denoising. In this setup, we corrupt the input images by adding Gaussian noise and proceed with computing their corresponding sparse codes through amortized inference using the encoder. We then feed these codes to the decoder to obtain reconstructions of the noisy inputs.

Figure 7 shows the reconstructions of MNIST test set images and of the same images corrupted with Gaussian noise with zero mean and standard deviations $\sigma = 1$ and $\sigma = 1.5$ using the SDL and VDL models in Figures 2b and 2d. We observe that both SDL and VDL models are able to produce reconstructions which are closer to samples from the manifold of training data than to the noisy inputs. This implies that the sparse coding systems have learned to detect features useful for the data they are trained on and cannot reconstruct random noise, as desired. SDL and VDL models trained on ImageNet patches are also robust to the addition of Gaussian noise to their inputs. This is evident in Figure 8 which contains reconstructions $\tilde{\boldsymbol{y}}$ and $\tilde{\boldsymbol{y}}_\sigma$ of

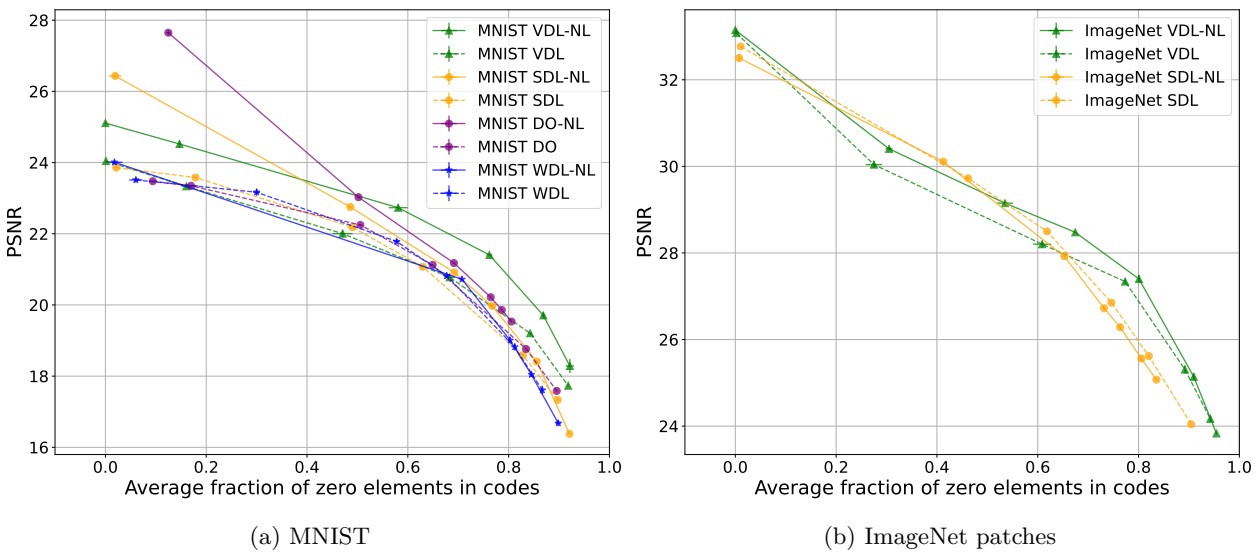

(a) MNIST

(b) ImageNet patches

Figure 6: Trade-off between sparsity measured by average percentage of inactive (zero) code components and reconstruction quality measured by PSNR for different models. Figure (6a) plots regret curves for SDL, SDL-NL, VDL, VDL-NL, WDL, WDL-NL, DO and DO-NL models trained on MNIST with code dimension $l = 128$ and different levels of sparsity regularization $\lambda$. Figure (6b) plots regret curves for SDL, SDL-NL, VDL, and VDL-NL models trained on ImageNet patches with code dimension $l = 256$ and different levels of sparsity regularization $\lambda$. Higher PSNR reflects better reconstruction quality.

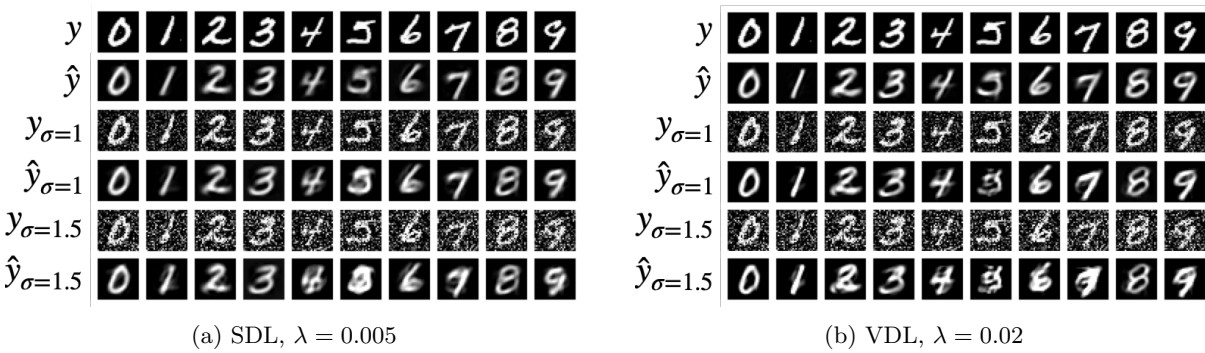

(a) SDL, $\lambda = 0.005$

(b) VDL, $\lambda = 0.02$

Figure 7: Examples of denoising abilities of encoders from SDL and VDL models trained on MNIST which produce codes with average sparsity of 89.7% and 91.8%, respectively. Top two rows: original inputs from the test set (top) and their reconstructions (bottom); middle two rows: the inputs corrupted with Gaussian noise with standard deviation $\sigma = 1$ (top) and their reconstructions (bottom), bottom two rows: the inputs corrupted with Gaussian noise with standard deviation $\sigma = 1.5$ (top) and their reconstructions (bottom).

images $\boldsymbol{y}$ and their noisy version $\boldsymbol{y}_\sigma$ for the SDL and VDL models from Figures 3a and 3c which produce codes with similar sparsity on average.

Table 1 shows a summary of the reconstruction quality of images with varying levels of Gaussian noise corruption when they are passed through SDL and VDL autoencoders. The results are evaluated on the test set over 5 random seeds. Both the SDL and VDL encoders are able to produce codes from corrupted inputs $\boldsymbol{y}_\sigma$ which, when passed through their corresponding decoders, produce outputs $\tilde{\boldsymbol{y}}_\sigma$ that are less noisy than the corrupted inputs. This observation is significant since the sparse autoencoders in our experiments are not explicitly trained to perform denoising.

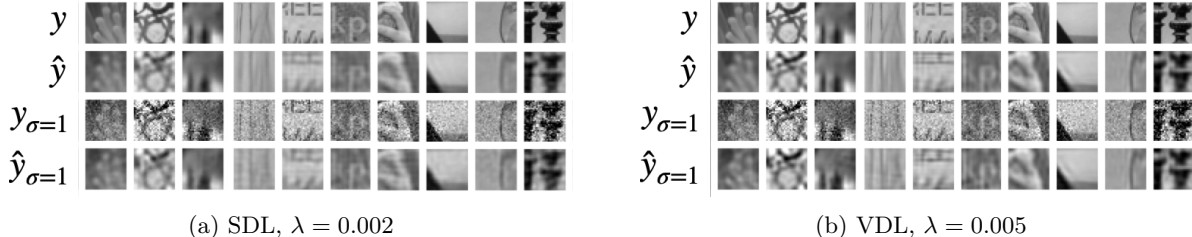

(a) SDL, $\lambda = 0.002$        (b) VDL, $\lambda = 0.005$

Figure 8: Examples of denoising abilities of encoders from SDL and VDL models trained on ImageNet patches which produce codes with average sparsity of 74.6% and 77.3%, respectively. Top two rows: original inputs from the test set (top) and their reconstructions (bottom); bottom two rows: the same inputs corrupted with Gaussian noise with standard deviation $\sigma = 1$ (top) and their reconstructions (bottom).

Table 1: The performance of SDL and VDL models trained on MNIST and ImageNet patches with different levels of sparsity regularization $\lambda$ on the the task of denoising inputs corrupted with Gaussian noise with zero mean and standard deviation $\sigma = 1.0$ or $\sigma = 1.5$. We report the average sparsity of codes computed with these models and the PSNR of: the reconstruction $\tilde{y}$ of the original input $y$, the corrupted input $y_\sigma$, and the reconstruction $\tilde{y}_\sigma$ of the corrupted input $y_\sigma$. Evaluated on the test set over 5 random seeds.

| | | | MNIST | | | | |
|---|---|---|---|---|---|---|---|
| MODEL | $\lambda$ | $L_0(z_{\tilde{y}})$ | PSNR $\tilde{y}$ | $\sigma$ | PSNR $y_\sigma$ | $L_0(z_{\tilde{y}_\sigma})$ | PSNR $\tilde{y}_\sigma$ |
| SDL | 0.005 | $89.7\% \pm 0.2\%$ | $17.3 \pm 0.028$ | 1.0 | 10.2 | $87.3\% \pm 0.2\%$ | $16.9 \pm 0.053$ |
| VDL | 0.02 | $91.8\% \pm 0.1\%$ | $17.7 \pm 0.023$ | 1.0 | 10.2 | $88.1\% \pm 0.2\%$ | $15.4 \pm 0.175$ |
| SDL | 0.005 | $89.7\% \pm 0.2\%$ | $17.3 \pm 0.028$ | 1.5 | 6.7 | $84.6\% \pm 0.5\%$ | $15.7 \pm 0.204$ |
| VDL | 0.02 | $91.8\% \pm 0.1\%$ | $17.7 \pm 0.023$ | 1.5 | 6.7 | $84.5\% \pm 0.3\%$ | $12.2 \pm 0.275$ |
| | | | IMAGENET PATCHES | | | | |
| SDL | 0.002 | $74.6\% \pm 0.1\%$ | $26.9 \pm 0.007$ | 1.0 | 20.0 | $72.7\% \pm 0.1\%$ | $25.6 \pm 0.011$ |
| VDL | 0.005 | $77.3\% \pm 0.2\%$ | $27.3 \pm 0.058$ | 1.0 | 20.0 | $73.5\% \pm 0.2\%$ | $25.2 \pm 0.035$ |

## 4.4 Ablation: No Variance Regularization

VDL and VDL-NL experiments from Figure 6 show that our proposed variance regularization strategy can successfully be used to train sparse coding models without collapse in the $l_1$ norms of the sparse codes. In experiments without our variance regularization strategy or any regularization restricting the dictionary's weights, we do observe collapse, both for models with a linear and a non-linear decoder. This shows that our proposed variance regularization protocol is an effective alternative to the standard $l_2$ normalization or weight decay regularization for preventing collapse, both in the case of models with a linear decoder and with a fully connected decoder with one hidden layer.

## 4.5 Classification in the Low Data Regime

We investigate whether self-supervised pre-training with sparse coding improves classification performance over training from scratch in the low data regime. For this purpose, we compare the classification error of classifiers trained on the raw MNIST data to the classification error of linear classifiers trained on features from pre-trained frozen LISTA encoders in SDL, SDL-NL, VDL, and VDL-NL autoencoders[6] and pre-trained frozen DO and DO-NL decoders when few training samples are available for supervised learning.

In particular, we train a linear classifier on top of features from these pre-trained models when 1, 2, 5, 10, 20, 50, 100 training samples per class are available as well as when the full dataset is available. For each type of pre-trained model and each number of training samples per class, we select the model among the

---

[6]The autoencoders are trained on the full MNIST dataset in an unsupervised way as described in section 1.

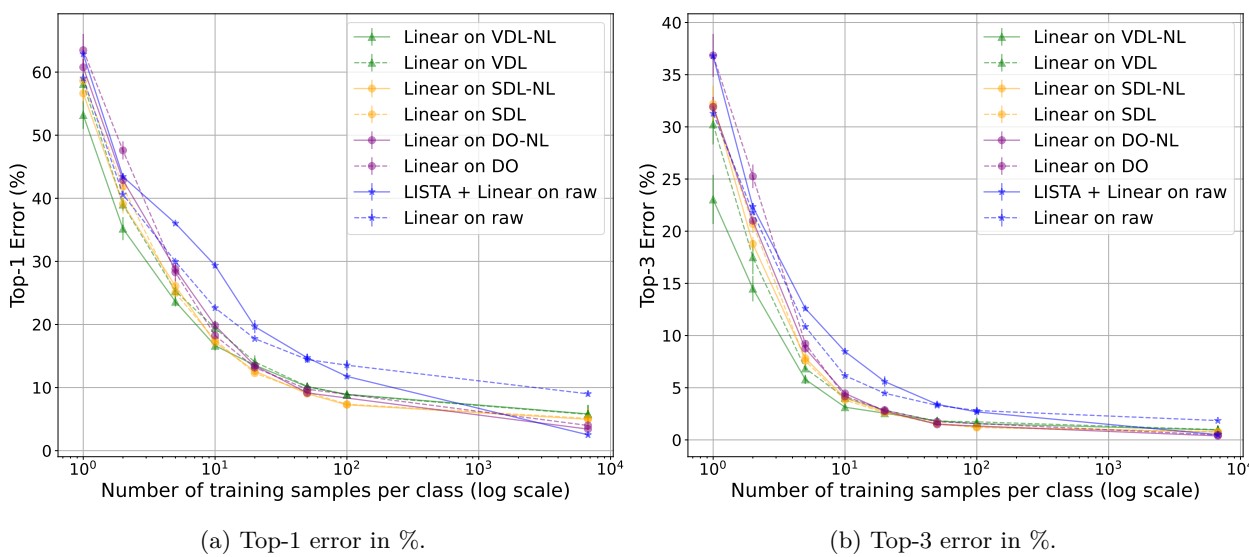

(a) Top-1 error in %.

(b) Top-3 error in %.

Figure 9: Linear separability (measured by classification error) of the raw MNIST data and of features from pre-trained SDL, SDL-NL, VDL, VDL-NL encoders and decoder-only DO and DO-NL models with different numbers of training samples per class. Additionally, classification error of a classifier consisting of a LISTA enocder followed by a linear layer (LISTA + Linear) trained from scratch on the raw MNIST data. Models trained with variance regularization in the codes and a non-linear decoder (VDL-NL) outperform other models in linear classification with up to 10 training samples per class. Test set performance is reported over 5 random seeds.

ones presented in Figure 6a which gives the best validation accuracy. As baselines, we train two classifiers from scratch on the raw MNSIT data. The first one is a linear classifier referred to as "Linear on raw". The second one consists of a linear layer on top of a randomly initialized LISTA encoder [7] and is referred to as "LSITA + Linear on raw".

Figure 9 summarizes the top-1 and top-3 classification errors of these classifiers on the test set evaluated over 5 random seeds. With small exceptions[8], all linear classifiers trained on top of the pre-trained LISTA features give better top-1 and top-3 classification performance than classifiers trained on the raw MNIST data with up to 100 samples per class. This observation supports a claim made by Raina et al. (2007) that SSL is expected to be useful especially when labeled data is scarce. Notably, classifiers trained on top of VDL-NL features have the best performance among the rest of the models with up to 10 training samples per class. In other words, sparse autoencoders with a non-linear decoder pre-trained using our proposed variance regularization on the latent codes can boost the classification performance when few labeled training samples are available.

## 5 Related Work

Our work is related to the extensive literature on sparse coding and dictionary learning. Mairal et al. (2014) provide a comprehensive overview of sparse modeling in computer vision, discuss its relation to the fields of statistics, information theory, and signal processing, as well as its applications in computer vision including image denoising, inpainting, and super-resolution.

**Ways to Avoid Collapse** There are different ways to avoid the collapse issue in a sparse coding model. One approach is to constrain the weights of the decoder, for example by bounding the norms of its columns (Lee et al., 2006; Raina et al., 2007; Mairal et al., 2009; Hu & Tan, 2018) or by applying weight decay regularization (Sun et al., 2018). Another approach is to regularize the latent codes directly - a strategy we

---

[7]This randomly initialized LISTA encoder is trainable and with the same architecture as the one in Algorithm 1.

[8]Namely, classifiers trained on DO, DO-NL and SDL features with 1 or 2 training samples per class.

incorporate in our model. In the spiking model of Földiak (1990), an anti-Hebbian adaptive thresholding mechanism encourages each latent component to be active with a pre-set level of probability. Our variance regularization strategy is also related to Olshausen & Field (1996; 1997) where the norm of the basis elements is adjusted adaptively over time so that each of the latent components has a desired level of variance. Their method works well in the linear dictionary setup but it is not directly extendable to the case of a non-linear decoder. In variational inference models such as VAE Kingma & Welling (2013) and its sparse coding extensions such as SVAE Barello et al. (2018) and Tonolini et al. (2020), the latent codes have a pre-set level of variance determined by the prior distribution. In SVAE, there is also variance regularization term which determines the trade-off between the input reconstruction and the sparsity level imposed by the prior. Our variance regularization term is most closely related to the variance principle presented in Bardes et al. (2021) in which each latent component is encouraged to have variance above a pre-set threshold.

**Sparse Coding and Classification Performance** There is an active area of research which studies whether sparse coding is useful in classification tasks. Some existing literature supports this claim Raina et al. (2007); Ranzato et al. (2007); Mairal et al. (2008a;b); Yang et al. (2009); Kavukcuoglu et al. (2010); Makhzani & Frey (2013); He et al. (2014) while other research refutes it Coates & Ng (2011); Rigamonti et al. (2011); Luther & Seung (2022). Tang et al. (2020) show that adding layers to a hierarchy of sparse dictionaries can improve classification accuracy. In our work, we provide empirical evidence that sparse coding models trained with a reconstruction objective, especially models with a non-linear decoder trained using our variance regularization method, can give better classification performance than classifiers trained on the raw data when very few training samples are available, supporting a claim made by Raina et al. (2007) that SSL is expected to be useful when labeled data is scarce.

**Sparsity Constraints** Sparse representations can be obtained using many different objectives. The $l_1$ penalty term in equation 1 can be replaced by other sparsity-inducing constraints including $l_0$, $l_p$ ($0 < p < 1$), $l_2$, or $l_{2,1}$ norms. In particular, the $l_1$ term is a convex relaxation of the non-differentiable $l_0$ constraint of the form $\|z\|_0 \leq k$ requiring $z$ to have at most $k$ active components. Zhang et al. (2015) provide a survey of existing algorithms for sparse coding such as K-SVD which generalizes K-means clustering (Aharon et al., 2006). Makhzani & Frey (2013) propose an approach to train a sparse autoencoder with a linear encoder and decoder in which only the top $k$ activations in the hidden layer are kept and no other sparsifying penalty is used. In our model, we modify the commonly used sparse coding setup with $l_1$ penalty term in 1 by including a term which encourages a pre-set level of variance across each latent component in 9.

**Inference** There are many variations of the ISTA inference algorithm which address the fact that it is computationally expensive and are designed to speed it up, such as FISTA (Beck & Teboulle, 2009) which is used in our setup (for details please refer to section 2.2). Gated LISTA (Wu et al., 2019) introduces a novel gated mechanism for LISTA (Gregor & LeCun, 2010) which enhances its performance. In Zhou et al. (2018), it is shown that deep learning with LSTMs can be used to improve learning sparse codes in ISTA by modeling the history of the iterative algorithm and acting like a momentum. Inspired by LISTA, we train an encoder to predict the non-negative representations computed during inference using the original and modified FISTA protocols as explained in section 2.4.

**Multi-layer Sparse Coding** There exist methods which greedily construct hierarchies of sparse representations. Zeiler et al. (2010) use ISTA in a top-down convolutional multi-layer architecture to learn sparse latent representations at each level in a greedy layer-wise fashion. Other convolutional models which produce a hierarchy of sparse representations are convolutional DBNs (Lee et al., 2009) and hierarchical convolutional factor analysis (Chen et al., 2013) which also use layer-wise training. Other greedy approaches are proposed by He et al. (2014) and Tariyal et al. (2016) who train multiple levels of dictionaries. Chen et al. (2018) combine sparse coding and manifold learning in a multi-layer network which is also trained layer-wise. In contrast, our work is about finding sparse representations by utilizing multi-layer decoders which are trained end-to-end. Hu & Tan (2018) propose a non-linear sparse coding method in which a linear dictionary is trained in the encoding space of a non-linear autoencoder but it differs from our method as the sparse representations of this encoding layer are stand-alone, i.e. they are not fed as inputs to the multi-layer decoder. Other multi-layer sparse coding models include Sun et al. (2018), Mahdizadehaghdam et al. (2019), and Tang et al. (2020) in which linear dictionaries are stacked at different scales. Rather than using linear dictionaries, our method learns a non-linear decoder to decode sparse codes.

# 6    Conclusion

In this work, we revisit the traditional setup of sparse coding with an $l_1$ sparsity penalty. We propose to apply variance regularization to the sparse codes during inference with the goal of training non-linear decoders without collapse in the $l_1$ norm of the codes. We show that using our proposed method we can successfully train sparse autoencoders with fully connected multi-layer decoders which have interpretable features, outperform models with linear decoders in terms of reconstruction quality for a given average level of sparsity in the codes, and improve MNIST classification performance in the low data regime. Future research directions include scaling up our method to natural images using deep convolutional decoders.

**Broader Impact Statement**

This work proposes a general-purpose machine learning algorithm for representation learning which is trained using large amounts of unlabeled data. The representations learned by this algorithm reflect any biases present in the training data. Therefore, practitioners should apply techniques for identifying and mitigating the biases in the data when applicable. This notice is doubly important since the learned representations can be incorporated in numerous real-world applications such as medical diagnosis, surveillance systems, autonomous driving, and so on. These applications can have both positive and negative effect on society, and warrant extensive testing before being deployed in order to prevent harm or malicious use.

**Acknowledgments**

We would like to thank our colleagues at the NYU CILVR Lab and NYU Center for Data Science for insightful discussions, and to the TMLR anonymous reviewers and action editor for their feedback. Part of this research was conducted while Katrina was interning at FAIR. This work is supported in part by AFOSR award FA9550-19-1-0343 and by the National Science Foundation under NSF Award 1922658.

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

## A  Appendix

### A.1  Notation

Table 2 contains descriptions of the notation and symbols used in this work.

### A.2  Variance Regularization Term

Figure 10 visualizes the variance regularization term $h(\boldsymbol{x}) = \left[\left(1 - \sqrt{\mathrm{Var}(\boldsymbol{x})}\right)_+\right]^2$ for $\boldsymbol{x} \in \mathbb{R}^2$ in 3D.

### A.3  Gradient Derivation

We compute the gradient of the hinge term in (9) with respect to one latent code's component $Z_{s,t}$:

$$\frac{\partial}{\partial Z_{s,t}} \sum_{j=1}^{l} \beta \left[\left(T - \sqrt{\mathrm{Var}(\boldsymbol{Z}_{j,:})}\right)_+\right]^2 \tag{16}$$

Table 2: Descriptions of symbols and notation used in the paper.

| Notation | Description |
| --- | --- |
| $\mathcal{E}$ | LISTA-inspired encoder. |
| $\mathcal{D}$ | Decoder: linear or fully connected with one hidden layer. |
| $m$ | Dimension of the decoder's hidden layer. |
| $\lambda$ | Hyperparameter for the sparsity regularization. |
| $\beta$ | Hyperparameter for the variance regularization of the latent codes. |
| $\gamma$ | Hyperparameter encouraging the FISTA codes to be close to the encoder's predictions. |
| $d$ | Dimension of the input data. |
| $\boldsymbol{y}$ | Input sample in $\mathbb{R}^d$. |
| $N$ | Total number of training samples. |
| $n$ | Mini-batch size. |
| $T$ | Number of ISTA/FISTA iterations. |
| $\eta_k$ | Step size for ISTA/FISTA gradient step. |
| $\eta_{\mathcal{D}}$ | Learning rate for the decoder. |
| $\eta_{\mathcal{E}}$ | Learning rate for the encoder. |
| $\tau_\alpha$ | Shrinkage function $[\tau_\alpha(\boldsymbol{x})]_j = \text{sign}(x_j)(|x_j| - \alpha)$. |
| $\tilde{\tau}_\alpha$ | Non-negative shrinkage function $[\tilde{\tau}_\alpha(\boldsymbol{x})]_j = ([\tau_\alpha(\boldsymbol{x})]_j)_+$. |
| $l$ | Dimension of the latent code. |
| $\boldsymbol{z}$ | Latent code in $\mathbb{R}^l$. |
| $\boldsymbol{Z}$ | Mini-batch of latent codes in $\mathbb{R}^{l \times n}$. |
| $\boldsymbol{W}$ | Parametrization in $\mathbb{R}^{d \times l}$ of a linear dictionary $\mathcal{D}$. |
| $\boldsymbol{W}_1$ | Parametrization in $\mathbb{R}^{m \times l}$ of the bottom layer of a fully connected decoder $\mathcal{D}$ with one hidden layer. |
| $\boldsymbol{W}_2$ | Parametrization of a the top layer of a fully connected decoder $\mathcal{D}$ with one hidden layer in $\mathbb{R}^{d \times m}$. |
| $\boldsymbol{b}_1$ | Parametrization of the bias term in $\mathbb{R}^m$ following the bottom layer of a fully connected decoder $\mathcal{D}$ with one hidden layer. |
| $\boldsymbol{b}$ | Parametrization of the bias term in the LISTA encoder (see 3.1). |
| SDL | Standard Dictionary Learning with a LISTA encoder and a linear decoder whose columns have a fixed $l_2$ norm. |
| SDL-NL | Standard Dictionary learning with a LISTA encoder and a non-linear fully connected decoder. Each layer in the decoder has columns with a fixed $l_2$ norm. |
| VDL | Variance-regularized Dictionary Learning with a LISTA encoder and a linear decoder in which regularization is applied to the sparse codes encouraging the variance across the latent components to be above a fixed threshold. |
| VDL-NL | Variance-regularized Dictionary Learning (as above) with a non-linear decoder. |
| WDL | Dictionary Learning with a LISTA encoder and a linear decoder trained with weight decay regularization. |
| WDL-NL | Dictionary Learning with a LISTA encoder and a non-linear decoder trained with weight decay regularization. |
| DO | Standard Dictionary Learning without an encoder and with a linear decoder whose columns have a fixed $l_2$ norm. |
| DO-NL | Standard Dictionary Learning without an encoder and with a non-linear decoder whose columns have a fixed $l_2$ norm. |

As a reminder of our notation, $l$ is the latent dimension, $n$ is the batch size, $\boldsymbol{Z} \in \mathbb{R}^{l \times n}$ stores the codes for all elements in a batch, and $\boldsymbol{Z}_{j,:} \in \mathbb{R}^n$ stores code values for the $j$-th latent component.

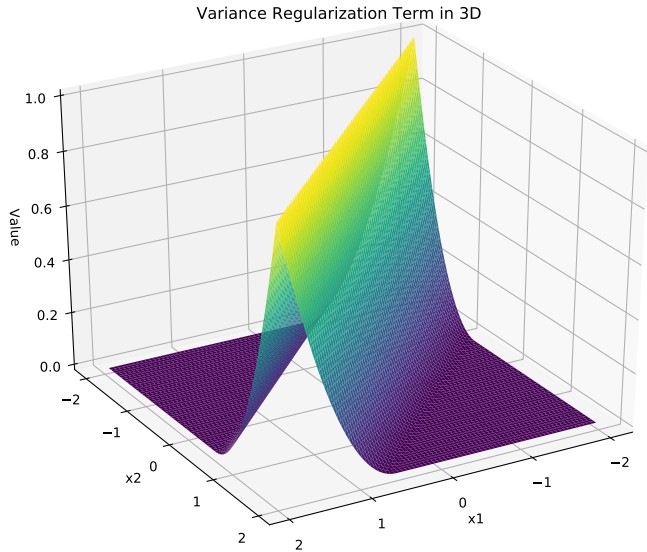

Figure 10: Visualizing the variance regularization function $h(\boldsymbol{x}) = \left[\left(1 - \sqrt{\mathrm{Var}(\boldsymbol{x})}\right)_+\right]^2$ for $\boldsymbol{x} \in \mathbb{R}^2$ in 3D.

Note that:

$$\mathrm{Var}(\boldsymbol{Z}_{j,:}) = \frac{1}{n-1} \sum_{i=1}^{n} (Z_{j,i} - \mu_j)^2, \tag{17}$$

$$\mu_j = \frac{1}{n} \sum_{i=1}^{n} Z_{j,i}. \tag{18}$$

The gradient in (16) is non-zero only for $j = s$ and $\sqrt{\mathrm{Var}(\boldsymbol{Z}_{s,:})} < T$. Thus,

$$\frac{\partial}{\partial Z_{s,t}} \sum_{j=1}^{l} \beta\left[\left(T - \sqrt{\mathrm{Var}(\boldsymbol{Z}_{j,:})}\right)_+\right]^2 = \frac{\partial}{\partial Z_{s,t}} \beta\left(T - \sqrt{\mathrm{Var}(\boldsymbol{Z}_{s,:})}\right)^2 \tag{19}$$

$$= 2\beta\left(T - \sqrt{\mathrm{Var}(\boldsymbol{Z}_{s,:})}\right) \frac{\partial}{\partial Z_{s,t}} \left(T - \sqrt{\mathrm{Var}(\boldsymbol{Z}_{s,:})}\right) \tag{20}$$

$$= -2\beta\left(T - \sqrt{\mathrm{Var}(\boldsymbol{Z}_{s,:})}\right) \frac{\partial}{\partial Z_{s,t}} \sqrt{\mathrm{Var}(\boldsymbol{Z}_{s,:})} \tag{21}$$

$$= -\beta \frac{\left(T - \sqrt{\mathrm{Var}(\boldsymbol{Z}_{s,:})}\right)}{\sqrt{\mathrm{Var}(\boldsymbol{Z}_{s,:})}} \frac{\partial}{\partial Z_{s,t}} \mathrm{Var}(\boldsymbol{Z}_{s,:}) \tag{22}$$

Now, continuing with:

$$\frac{\partial}{\partial Z_{s,t}} \mathrm{Var}(\boldsymbol{Z}_{s,:}) = \frac{\partial}{\partial Z_{s,t}} \left[\frac{1}{n-1} \sum_{i=1}^{n} (Z_{s,i} - \mu_s)^2\right] \tag{23}$$

$$= \frac{1}{n-1}\Big[\frac{\partial}{\partial Z_{s,t}}(Z_{s,t}-\mu_s)^2 + \sum_{i\neq t}\frac{\partial}{\partial Z_{s,t}}(Z_{s,i}-\mu_s)^2\Big] \tag{24}$$

$$= \frac{1}{n-1}\Big[2(Z_{s,t}-\mu_s)\frac{\partial}{\partial Z_{s,t}}(Z_{s,t}-\mu_s) + \sum_{i\neq t}2(Z_{s,i}-\mu_s)\frac{\partial}{\partial Z_{s,t}}(Z_{s,i}-\mu_s)\Big] \tag{25}$$

$$= \frac{2}{n-1}\Big[(Z_{s,t}-\mu_s)\frac{\partial}{\partial Z_{s,t}}\Big(Z_{s,t}-\frac{1}{n}\sum_{m=1}^{n}Z_{s,m}\Big) + \sum_{i\neq t}(Z_{s,i}-\mu_s)\frac{\partial}{\partial Z_{s,t}}\Big(Z_{s,i}-\frac{1}{n}\sum_{m=1}^{n}Z_{s,m}\Big)\Big] \tag{26}$$

$$= \frac{2}{n-1}\Big[(Z_{s,t}-\mu_s)\Big(1-\frac{1}{n}\Big) + \sum_{i\neq t}(Z_{s,i}-\mu_s)\Big(-\frac{1}{n}\Big)\Big] \tag{27}$$

$$= \frac{2}{n-1}\Big[(Z_{s,t}-\mu_s) - \frac{1}{n}\sum_{i=1}^{n}(Z_{s,i}-\mu_s)\Big] \tag{28}$$

$$= \frac{2}{n-1}\Big[(Z_{s,t}-\mu_s) - \frac{1}{n}\sum_{i=1}^{n}Z_{s,i} + \frac{1}{n}n\mu_s\Big] \tag{29}$$

$$= \frac{2}{n-1}\Big[(Z_{s,t}-\mu_s) - \mu_s + \mu_s\Big] \tag{30}$$

$$= \frac{2}{n-1}(Z_{s,t}-\mu_s), \tag{31}$$

and using the result in (22) we conclude that:

$$\frac{\partial}{\partial Z_{s,t}}\sum_{j=1}^{l}\beta\Big[\Big(T-\sqrt{\mathrm{Var}(\boldsymbol{Z}_{j,:})}\Big)_+\Big]^2 = \begin{cases} -\frac{2\beta}{n-1}\frac{\Big(T-\sqrt{\mathrm{Var}(\boldsymbol{Z}_{s,:})}\Big)}{\sqrt{\mathrm{Var}(\boldsymbol{Z}_{s,:})}}(Z_{s,t}-\mu_s) & \text{if } \sqrt{\mathrm{Var}(\boldsymbol{Z}_{s,:})} < T \\ 0 & \text{otherwise.} \end{cases} \tag{32}$$

## B   Additional Visualizations

Figures 11 and 13 display 128 of the dictionary elements for the same SDL and VDL models as in Figures 2 and 3. Figure 12 displays 128 of the dictionary elements for the WDL and DO models trained with different levels of sparsity regularization.

### B.1   Training Details

All hyperparameter values are selected through grid search. For the constant step size $\eta_k$ in FISTA, we consider values in the range of 0.01 to 100. In VDL experiments, we consider values for the hinge regularization coefficient $\beta$ between 1e−1 and 100. In all our experiments, we use Adam (Kingma & Ba, 2014) as an optimizer for both the encoder $\mathcal{E}$ and decoder $\mathcal{D}$ with a batch size of 250. We use $L = 3$ iterations in the LISTA encoder (see Algorithm 1). We run the FISTA algorithm for a maximum of 200 iterations. The convergence criterion we set is: $\frac{\|\boldsymbol{z}^{(k)} - \boldsymbol{z}^{(k-1)}\|_2}{\|\boldsymbol{z}^{(k-1)}\|_2} < 1\text{e}{-}3$. FISTA's output is $\boldsymbol{z}^* = \boldsymbol{z}^{(k^*)}$ where $k^*$ is the index of the first iteration for which the convergence criterion holds. Table 3 contains the hyperparameter values we use in all our experiments expect for the ones with WDL, WDL-NL, DO and DO-NL models. In the case of DO and DO-NL models, we use the same number of epochs, $\eta_{\mathcal{D}}$, wd($\boldsymbol{b}$), wd($\boldsymbol{b}_1$), and $\eta_k$ as in SDL and SDL-NL experiments, respectively. In the case of WDL and WDL-NL models, we use grid search to find the optimal weight decay value of 5e−4 (in terms of reconstruction quality regret curve as displayed in Figure 6; this is also the value used in Sun et al. (2018)) and use the same number of epochs, $\eta_{\mathcal{D}}$, $\gamma$, $\eta_{\mathcal{E}}$, and $\eta_k$ as in SDL and SDL-NL experiments, respectively.

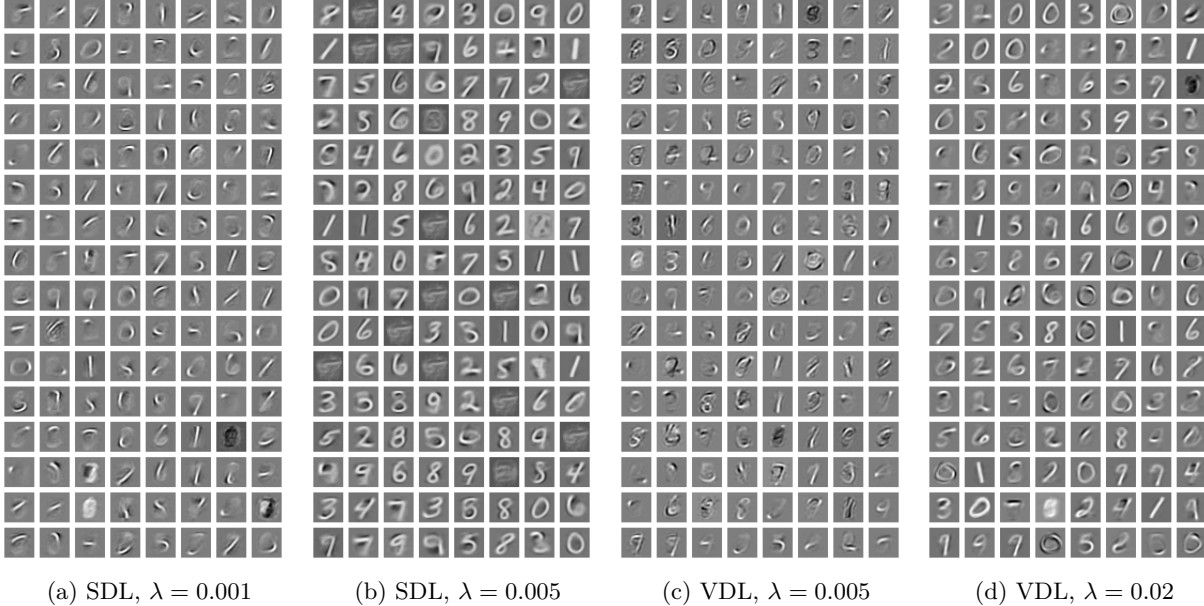

(a) SDL, $\lambda = 0.001$     (b) SDL, $\lambda = 0.005$     (c) VDL, $\lambda = 0.005$     (d) VDL, $\lambda = 0.02$

Figure 11: Dictionary elements for linear dictionaries trained on MNIST with latent dimension $l = 128$ and different levels of sparsity regularization $\lambda$. SDL stands for dictionary learning in which the $l_2$ norm of the decoder's columns is fixed to 1. VDL stands for dictionary learning in which norms of the decoder's columns are not explicitly bounded but our proposed variance regularization on the latent codes is used. The SDL models in 2a and 2b produce reconstructions with average PSNR of 21.1 and 18.6 for codes with average sparsity level of 63% and 83% on the test set, respectively. The VDL models in 2c and 2d produce reconstructions with average PSNR of 20.7 and 17.7 for codes with average sparsity level of 69% and 91.8% on the test set, respectively.

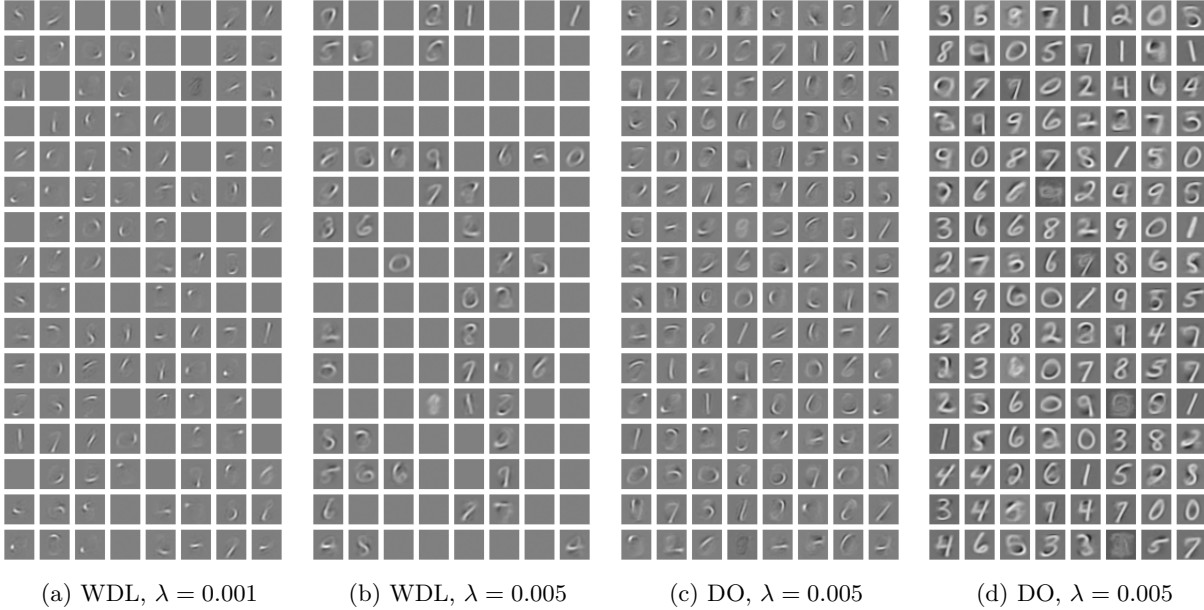

(a) WDL, $\lambda = 0.001$     (b) WDL, $\lambda = 0.005$     (c) DO, $\lambda = 0.005$     (d) DO, $\lambda = 0.005$

Figure 12: Dictionary elements for linear dictionaries trained on MNIST with latent dimension $l = 128$ and different levels of sparsity regularization $\lambda$. WDL stands for dictionary learning in which weight decay is applied to the decoder's weights and there is a LISTA encoder. DO stands for dictionary learning in which norms of the decoder's columns are fixed to 1 and there is no LISTA encoder. The WDL models in 12a and 12b produce reconstructions with average PSNR of 20.8 and 17.6 for codes with average sparsity level of 67.7% and 86.7% on the test set, respectively. The DO models in 12c and 12d produce reconstructions with average PSNR of 21.1 and 17.6 for codes with average sparsity level of 64.9% and 89.5% on the test set, respectively.

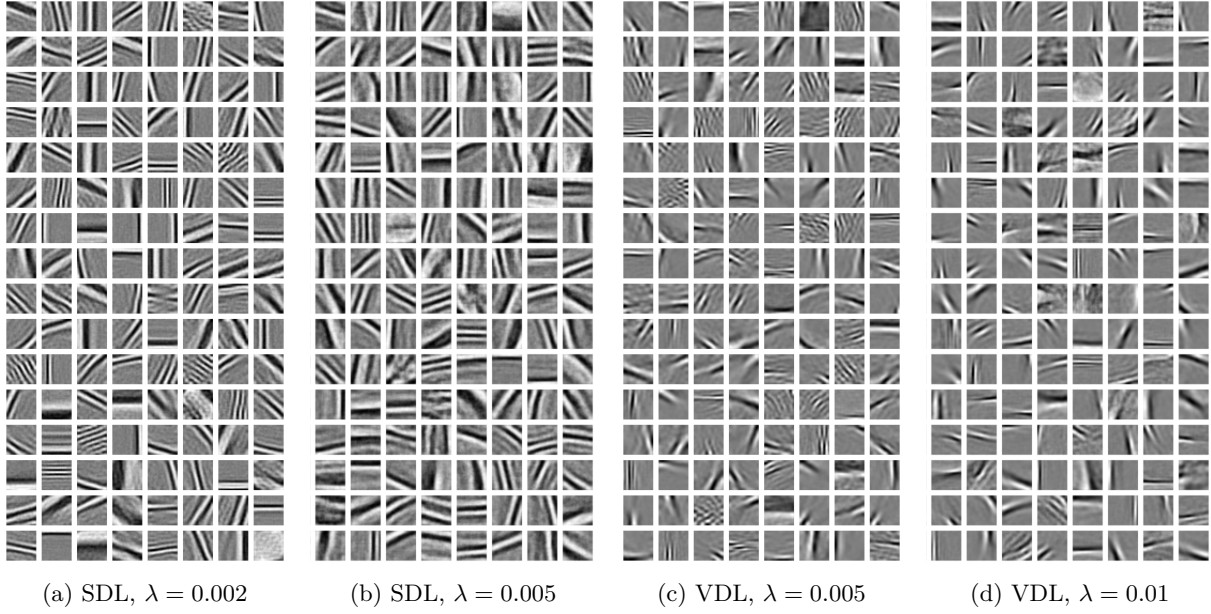

(a) SDL, $\lambda = 0.002$      (b) SDL, $\lambda = 0.005$      (c) VDL, $\lambda = 0.005$      (d) VDL, $\lambda = 0.01$

Figure 13: Dictionary elements which resemble Gabor filters for SDL and VDL models with latent dimension $l = 256$ trained on ImageNet patches with different levels of sparsity regularization $\lambda$. The SDL models in 3a and 3b produce reconstructions with average PSNR of 26.9 and 24.0 and average sparsity level in the codes of 74.6% and 90.5% on the test set, respectively. The VDL models in Figures 3c and 3d produce reconstructions with average PSNR of 27.3 and 25.3 and average sparsity level in the codes of 77.3% and 89.2% on the test set, respectively.

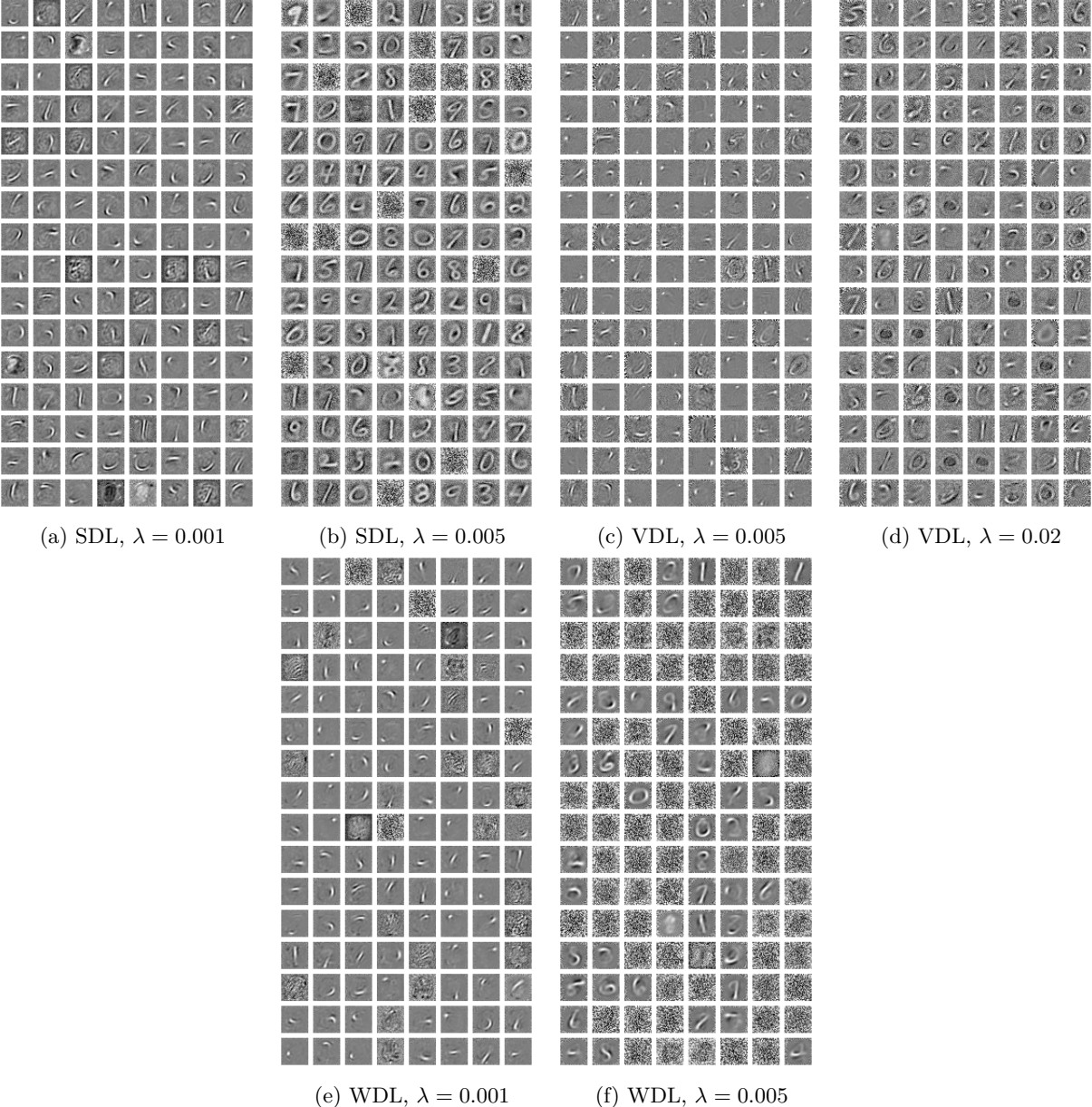

(a) SDL, $\lambda = 0.001$     (b) SDL, $\lambda = 0.005$     (c) VDL, $\lambda = 0.005$     (d) VDL, $\lambda = 0.02$

(e) WDL, $\lambda = 0.001$     (f) WDL, $\lambda = 0.005$

Figure 14: Visualization of the $\boldsymbol{U} \in \mathbb{R}^{d \times l}$ layer of the LISTA encoder for SDL, VDL, and WDL models with code dimension $l = 128$ trained on MNIST images and different levels of sparsity regularization $\lambda$. Similarly to the decoders in 11, the features learned by the encoders resemble orientations and parts of digits but in the case of WDL models there are more noisy features compared to the other models as $\lambda$ increases.

| dataset | model | ep | $\lambda$ | $\gamma$ | $\beta$ | $T$ | $\eta_{\mathcal{D}}$ | $\eta_{\mathcal{E}}$ | wd($\boldsymbol{b}_1$) | wd($\boldsymbol{b}$) | $\eta_k$ |
|---|---|---|---|---|---|---|---|---|---|---|---|
| MNIST | SDL | 200 | 0 | 1 | 0 | - | 1e−3 | 3e−4 | - | 0 | 1 |
| MNIST | SDL | 200 | 1e−4 | 1 | 0 | - | 1e−3 | 3e−4 | - | 0 | 1 |
| MNIST | SDL | 200 | 5e−4 | 1 | 0 | - | 1e−3 | 3e−4 | - | 0 | 1 |
| MNIST | SDL | 200 | 1e−3 | 1 | 0 | - | 1e−3 | 3e−4 | - | 0 | 1 |
| MNIST | SDL | 200 | 3e−3 | 1 | 0 | - | 1e−3 | 3e−4 | - | 0 | 1 |
| MNIST | SDL | 200 | 5e−3 | 1 | 0 | - | 1e−3 | 3e−4 | - | 0 | 1 |
| MNIST | VDL | 200 | 0 | 5 | 10 | 0.5 | 3e−4 | 1e−4 | - | 0 | 0.5 |
| MNIST | VDL | 200 | 1e−3 | 5 | 10 | 0.5 | 3e−4 | 1e−4 | - | 0 | 0.5 |
| MNIST | VDL | 200 | 3e−3 | 5 | 10 | 0.5 | 3e−4 | 1e−4 | - | 0 | 0.5 |
| MNIST | VDL | 200 | 5e−3 | 5 | 10 | 0.5 | 3e−4 | 1e−4 | - | 0 | 0.5 |
| MNIST | VDL | 200 | 1e−2 | 5 | 10 | 0.5 | 3e−4 | 1e−4 | - | 0 | 0.5 |
| MNIST | VDL | 200 | 2e−2 | 5 | 10 | 0.5 | 3e−4 | 1e−4 | - | 0 | 0.5 |
| MNIST | SDL-NL | 200 | 0 | 1 | 0 | - | 1e−3 | 1e−4 | 1e−3 | 0 | 1 |
| MNIST | SDL-NL | 200 | 1e−3 | 1 | 0 | - | 1e−3 | 1e−4 | 1e−3 | 0 | 1 |
| MNIST | SDL-NL | 200 | 3e−3 | 1 | 0 | - | 1e−3 | 1e−4 | 1e−3 | 0 | 1 |
| MNIST | SDL-NL | 200 | 5e−3 | 1 | 0 | - | 1e−3 | 1e−4 | 1e−3 | 0 | 1 |
| MNIST | SDL-NL | 200 | 1e−2 | 1 | 0 | - | 1e−3 | 1e−4 | 1e−3 | 0 | 1 |
| MNIST | SDL-NL | 200 | 2e−2 | 1 | 0 | - | 1e−3 | 1e−4 | 1e−3 | 0 | 1 |
| MNIST | VDL-NL | 200 | 0 | 100 | 10 | 0.5 | 3e−4* | 1e−4 | 1e−3 | 0 | 0.5 |
| MNIST | VDL-NL | 200 | 1e−3 | 100 | 10 | 0.5 | 3e−4* | 1e−4 | 1e−3 | 0 | 0.5 |
| MNIST | VDL-NL | 200 | 3e−3 | 100 | 10 | 0.5 | 3e−4* | 1e−4 | 1e−3 | 0 | 0.5 |
| MNIST | VDL-NL | 200 | 5e−3 | 100 | 10 | 0.5 | 3e−4* | 1e−4 | 1e−3 | 0 | 0.5 |
| MNIST | VDL-NL | 200 | 1e−2 | 100 | 10 | 0.5 | 3e−4* | 1e−4 | 1e−3 | 0 | 0.5 |
| MNIST | VDL-NL | 200 | 2e−2 | 100 | 10 | 0.5 | 3e−4* | 1e−4 | 1e−3 | 0 | 0.5 |
| ImageNet | SDL | 100 | 0 | 1 | 0 | - | 1e−3 | 1e−4 | - | 1e−2 | 0.5 |
| ImageNet | SDL | 100 | 5e−4 | 1 | 0 | - | 1e−3 | 1e−4 | - | 1e−2 | 0.5 |
| ImageNet | SDL | 100 | 1e−3 | 1 | 0 | - | 1e−3 | 1e−4 | - | 1e−2 | 0.5 |
| ImageNet | SDL | 100 | 2e−3 | 1 | 0 | - | 1e−3 | 1e−4 | - | 1e−2 | 0.5 |
| ImageNet | SDL | 100 | 3e−3 | 1 | 0 | - | 1e−3 | 1e−4 | - | 1e−2 | 0.5 |
| ImageNet | SDL | 100 | 5e−3 | 1 | 0 | - | 1e−3 | 1e−4 | - | 1e−2 | 0.5 |
| ImageNet | VDL | 100 | 0 | 5 | 10 | 0.5 | 3e−4 | 1e−4 | - | 1e−2 | 0.5 |
| ImageNet | VDL | 100 | 1e−3 | 5 | 10 | 0.5 | 3e−4 | 1e−4 | - | 1e−2 | 0.5 |
| ImageNet | VDL | 100 | 3e−3 | 5 | 10 | 0.5 | 3e−4 | 1e−4 | - | 1e−2 | 0.5 |
| ImageNet | VDL | 100 | 5e−3 | 5 | 10 | 0.5 | 3e−4 | 1e−4 | - | 1e−2 | 0.5 |
| ImageNet | VDL | 100 | 1e−2 | 5 | 10 | 0.5 | 3e−4 | 1e−4 | - | 1e−2 | 0.5 |
| ImageNet | VDL | 100 | 1.5e−2 | 5 | 10 | 0.5 | 3e−4 | 1e−4 | - | 1e−2 | 0.5 |
| ImageNet | SDL-NL | 100 | 0 | 1 | 0 | - | 1e−3 | 1e−4 | 1e−2 | 1e−2 | 0.5 |
| ImageNet | SDL-NL | 100 | 1e−3 | 1 | 0 | - | 1e−3 | 1e−4 | 1e−2 | 1e−2 | 0.5 |
| ImageNet | SDL-NL | 100 | 3e−3 | 1 | 0 | - | 1e−3 | 1e−4 | 1e−2 | 1e−2 | 0.5 |
| ImageNet | SDL-NL | 100 | 5e−3 | 1 | 0 | - | 1e−3 | 1e−4 | 1e−2 | 1e−2 | 0.5 |
| ImageNet | SDL-NL | 100 | 8e−3 | 1 | 0 | - | 1e−3 | 1e−4 | 1e−2 | 1e−2 | 0.5 |
| ImageNet | SDL-NL | 100 | 1e−2 | 1 | 0 | - | 1e−3 | 1e−4 | 1e−2 | 1e−2 | 0.5 |
| ImageNet | VDL-NL | 100 | 0 | 20 | 10 | 0.5 | 5e−5* | 1e−4 | 1e−1 | 1e−2 | 0.5 |
| ImageNet | VDL-NL | 100 | 1e−3 | 20 | 10 | 0.5 | 5e−5* | 1e−4 | 1e−1 | 1e−2 | 0.5 |
| ImageNet | VDL-NL | 100 | 2e−3 | 20 | 10 | 0.5 | 5e−5* | 1e−4 | 1e−1 | 1e−2 | 0.5 |
| ImageNet | VDL-NL | 100 | 3e−3 | 20 | 10 | 0.5 | 5e−5* | 1e−4 | 1e−1 | 1e−2 | 0.5 |
| ImageNet | VDL-NL | 100 | 5e−3 | 20 | 10 | 0.5 | 5e−5* | 1e−4 | 1e−1 | 1e−2 | 0.5 |
| ImageNet | VDL-NL | 100 | 1e−2 | 40 | 10 | 0.5 | 5e−5* | 1e−4 | 1e−1 | 1e−2 | 0.5 |
| ImageNet | VDL-NL | 100 | 2e−2 | 40 | 10 | 0.5 | 5e−5* | 1e−4 | 1e−1 | 1e−2 | 0.5 |

Table 3: Training hyperparameters. SDL indicates models in which columns of the decoder's layers have a fixed norm of 1. VDL indicates models in which variance regularization is applied to the codes during inference. (*) means that the learning rate for the decoder $\eta_{\mathcal{D}}$ is annealed by half every 30 epochs. wd stands for weight decay.

