# OpenReview forum: "Sparse Coding with Multi-layer Decoders using Variance Regularization"
_TMLR — Accepted by TMLR_

### Review · Reviewer_unXH · 2022-04-28

**Summary Of Contributions:**

The paper suggests a regularization that prevents the estimated sparse codes to collapse. The idea is to ensure that the sparse codes will have variance above a pre-defined threshold. This regularization can be used for learning data-adaptive representations via dictionary learning / auto-encoder. In contrast with standard dictionary learning schemes (e.g., KSVD), there is no need to normalize the dictionary atoms in an L2 sense, but there are additional hyperparameters to tune.

The authors train the entire dictionary learning pipeline using end-to-end learning, by unfolding FISTA iterations. The sparse code variance regularization is shown to yield comparable reconstructions to that of the standard dictionary atom normalization.

**Broader Impact Concerns:**

The authors explain the ethical implications clearly.

**Requested Changes:**

Please address points (3)-(7) above. Also, I believe that it will be great if the authors could discuss and clarify points (1)-(2).




**Strengths And Weaknesses:**

I enjoyed reading the paper! The idea of regularizing the sparse codes is new and has shown to lead to comparable reconstructions with standard normalization.

It will be great if the authors could clarify the following points:

1. I understand that the specific use the variance regularization is new, but the question that I am asking myself is this: why variance regularization is preferred over the use of weight decay (L2 regularization over the weights) or standard normalization? All those methods will avoid the code collapse issue, why one is better than the other?

2. In other words, is there any theoretical motivation that flashes out why variance regularization is preferred over the standard L2 norm normalization?

3. Moreover, in the new approach, there are two additional hyper-parameters to tune (beta and T) compared to SDL, and one additional hyper-parameter compared to weight decay. Why should one prefer to increase the number of hyperparameters and use the variance regularizer if the performance is similar to standard methods?

4. Weight decay is widely used in end-to-end multi-layer dictionary learning. I believe it will valuable to include a comparison with weight decay, as the latter also avoids the sparse codes to collapse.

Other, minor issues:

5. What is the effect of T and beta: how sensitive is the learning to the choice of this pair?

6. How should one tune the hyper-parameters?

7. I do not understand the “Non-Linear Decoder” architecture. Please include a mathematical/algorithmic description of this part.

---

> ### Author Response · Authors · 2022-05-20
> **Response to Reviewer unXH [R1]**
>
> We thank the reviewer for the thoughtful comments and suggestions! We are glad that they find the idea of regularizing the sparse codes novel. Below we summarize and address the questions and issues raised by the reviewer.
>
> **Q1-Q2. [R1] Why should variance regularization be preferred over standard normalization or L2 weight decay? Is there a theoretical motivation on why variance regularization should be preferred over standard normalization?**
>
> Variance regularization applied during training avoids the collapse issue **directly** (by ensuring that the variance across each latent code component for a batch of images is above a fixed threshold) whereas standard $l_2$-normalization of the dictionary elements and weight decay prevent collapse **indirectly**. Additionally, in the case of a multi-layer decoder and standard $l_2$-normalization of the columns of each layer, the model might be harder to optimize.
>
> Moreover, experimental results in the paper (e.g. in Fig. 4 which shows level of sparsity vs reconstruction quality and in Fig. 9 which shows classification accuracy in the low data regime) suggest that models that use variance regularization (especially VDL-NL models) perform favorably compared to models trained with the standard normalization approach (SDL, SDL-NL).
>
> **Q3. [R1] Variance regularization has more hyperparameters than standard normalization or weight decay - why should it be preferred?**
>
> The additional parameters in the variance regularization setup are $T$ (fixed threshold for the variance of the latent components) and $\beta$ - the weight of the variance regularization term. The value of $T$ can be arbitrarily set to some fixed value (in the same way the norm of the dictionary atoms in the standard $l_2$-normalization regime can be set to any value as long as it is constant). Once fixed, the value of $\beta$ can be determined in a sense similar to setting a Lagrange multiplier - it should be big enough so the constraints (having variance above a certain level in the latent components) are met.
>
> **Q4. [R1] Comparison with weight decay end-to-end multi-layer dictionary learning.**
>
> At this moment, we are not familiar with end-to-end unsupervised multi-layer sparse coding methods which rely solely on weight decay as a way to prevent collapse (in our search, we came across Sun, X., Nasrabadi, N.M. and Tran, T.D. (2018) ”Supervised deep sparse coding networks.” which uses weight decay but their objective is supervised). Would the reviewer please refer us to the related work on end-to-end multi-layer dictionary learning with weight decay they have in mind?
>
> **Q5-Q6. [R1] How sensitive is the learning to $T$ and beta? How should the hyper-parameters be tuned?**
>
> As mentioned in Q3 above, the value of $T$ can be set arbitrarily and beta can be adjusted accordingly to make sure it's big enough so that the variance regularization term becomes very small. Across all our experiments, we fixed $T$ to the value of 0.5 and finding $\beta$ did not require a lot of tuning - we performed grid search with several values in the range of 0 to 100 and picked one that makes the value of the variance regularization term to become miniscule during inference. Similarly, for $\gamma$ (the term encouraging FISTA outputs to be close to the encoder’s predictions) we performed grid search with several values in the range of 0 to 100 and picked one that makes the value of the encoder regularization term to become miniscule during inference.
>
> **Q7. [R1] What is the non-linear decoder architecture?**
>
> The non-linear decoder’s architecture is a fully connected neural network with one hidden layer and a ReLU non-linearity. The non-linear decoder $\mathcal{D}: \mathbb{R}^l \to \mathbb{R}^{d}$ is parameterized by two fully-connected layers $W_1 \in \mathbb{R}^{m\times l}$ and $W_2 \in \mathbb{R}^{d\times m}$ and a bias term $b_1\in\mathbb{R}^{m}$ and given an input code $z\in\mathbb{R}^l$, the decoder’s output is computed as follows: $\tilde{y} = W_2(W_1z + b_1)$ where $\tilde{y}\in\mathbb{R}^{d}$. We will clarify this in the paper.

---

> > ### Comment · Reviewer_unXH · 2022-06-02
> > **Follow-up**
> >
> > Thanks for your detailed response! You have addressed most of my comments. Below, I discuss the remaining issues.
> >
> > **Weight decay**
> >
> > I find variance normalization as an interesting way to avoid the collapse problem in sparse coding setups. While the authors compared VDL to SDL both in the linear and nonlinear setting, which is great, I believe that the addition of a baseline that uses weight decay to avoid the collapse problem will improve the paper. This is the first idea that comes up to mind for tackling this issue. And yes, the paper "Supervised deep sparse coding networks" was the one I was referring to. I am sorry for not providing a reference. In any case, to quote the authors of that paper: "to prevent the L2-norm of the dictionary to be arbitrarily large and recovering trivial sparse codes, we introduce regularizer $||D||_2^F$."
> >
> > **Requested change**
> >
> > Show on one experiment (or more if you prefer) the advantage of the 'variance regularization' over 'weight decay regularization'. I believe that this will improve the quality of your work.

---

> > > ### Author Response · Authors · 2022-06-21
> > > **Follow-up**
> > >
> > > We thank reviewer unXH [R1] for following up and providing ways to improve our work! We agree that weight decay is a valuable baseline to include in the paper.
> > >
> > > We would like to share that we have provided additional experimental results that compare models trained using our variance regularization method to models trained using weight decay regularization in terms of the tradeoff between sparsity level in the codes and the quality of their corresponding reconstructions. Based on this metric, the models with a non-linear decoder trained with variance regularization outperform models trained using variance regularization. The details can be found in the "Additional Experiments" post and the supplementary materials.
> > >
> > > We are grateful for the reviewer’s time and effort!

---

### Review · Reviewer_G1LD · 2022-05-05

**Summary Of Contributions:**

The authors propose a modification to the original sparse coding model that replaces the dependency of normalized dictionary weights with a latent code variance regularization term. They argue that this allows them to avoid mode collapse resulting from the existing l1 regularization. They additionally extend both the classic and their modified sparse coding model with a non-linear decoding network. They claim that these modifications improve reconstruction for a given sparsity level, denoising performance, and few-shot classification scores over a standard sparse coding-based classifier, while maintaining interpretability.

**Broader Impact Concerns:**

No ethical concerns.

**Requested Changes:**

I have asked that the authors provide additional context for how their work fits into the broader field. I have also asked that they perform additional ablation experiments to understand how each of their contributions impact performance.

**Strengths And Weaknesses:**

**Strengths**

The paper is well written in general. The descriptions of the sparse coding, ISTA, and FISTA algorithms are succinct and complete. I’m also glad to see that they extended their experiments beyond the typical reconstruction/denoising regime into classification. The provided gradient calculations and weight visualizations in the appendix are appreciated – they give a more complete understanding of what the model learned and reinforce their argument of interpretability.

**Weaknesses**

I have two primary concerns with the submission: first is a lack of research into related work in this area and second is the depth of experimentation. I will discuss each of these in turn below. Together they trace one underlying request: for the authors to take more care in providing appropriate context for how their results fit into the broader field. They should include a more thorough background investigation, provide additional comparisons to previous attempts, and conduct additional experiments investigating the effects of their own modifications.

**Related work**

The idea of using latent variance to regularize sparse codes enforcing a fixed dictionary norm is not new. In fact, one of the first sparse coding methods introduced by Foldiak has this term explicitly [1], which was extended to a learned model in [2]. This was extended further by [3] to produce a modern variant that does not use backpropagation for learning. However, it was recently shown to learn similar image features to more traditional sparse coding models by [4]. In early work from Olshausen & Field [5], the normalization of the weights was actually set to match the variance of the sparse codes, not scaled to a constant norm as the authors suggest (they note “The vector length (gain) of each basis function was adapted over time so as to maintain equal variance on each coefficient.”, although I admit that the practice of normalizing the dictionary columns to constant norm is much more common today). While somewhat orthogonal, the more modern SVAE model proposed by [6] also includes a variance-based adaptation scheme in the weight update rule and the inference method has constrained latent variance by definition due to the VAE framework. Some works focus on normalizing the weights to control the variance, while others focus on normalizing the variance itself. Both approaches are valid and provide core insights that give necessary context for assessing the novelty of the submitted work.

One of the core research areas that the authors are addressing can be summed up by their phrasing: “Sparse representations of images have also been shown to improve classification performance.” However, they lack references to a large volume of works in this space, leading me to assume that they are not aware of the many previous attempts to test this hypothesis. What’s more, there is active research in the area that both supports and refutes this claim. Examples of the former can be found from a number of papers [7-12]. Studies claiming to refute the benefit of sparse coding models also provide compelling counter-points that should be considered in this work, for example [13-15].

Minor note - The authors note that SSL has been achieved by “enforcing equivariant representations across transformations of the same image”. It might be of interest to readers if they also included examples more closely related to the sparse coding literature, such as [16-19], the first of which is also a hierarchical sparse coding model with a non-linear decoder.

**Summary**
I am not suggesting that it is necessary for the authors to provide an extensive background of “unsupervised learning” or “sparse coding”. However, their choice of breadth of citations will matter for context and assessing novelty. The citation list I’ve provided is far from exhaustive, and some may be less relevant than others. My suggestion is that the authors take some more time to consider how their work fits into the greater context of this area of research, and perform their own literature review in the context of the works I’ve named.

**2) Depth of experimentation**

The authors propose several modifications to the classic sparse coding setup: a non-linear decoder, a variance regularization term, a learned inference network (LISTA), and a modification to iterative inference (eq 13).

To ensure that I understand correctly, I will outline each of the models tested & the primary experiments:

SDL is a standard sparse coding setup. It uses the unmodified FISTA algorithm for inference (i.e. not including the extra term from eq 13), a linear decoder, and the decoder weights are projected to have an l2 norm of 1 after each weight update. SDL-NL is the same as SDL, but with a two-layer nonlinear decoder, where the columns of both layer weights are fixed to have an l2 norm of 1.

VDL is the proposed variance-regularized model. During training, a modified FISTA is used for inference, which includes the variance regularization term and a new LISTA regularization term (eq 13). During testing, LISTA is used for inference. The decoder weight norms are unconstrained. VDL-NL is the same as VDL, but with a two-layer nonlinear decoder with weights that have no normalization constraints.


For the denoising experiments, they provide corrupted images as inputs to the networks and measure the PSNR of the reconstructions.

For the few-shot classification experiments, all four networks are trained without supervision on the training set, then the weights are frozen, then a linear classifier is trained with some limited number of labeled samples.

Assuming these descriptions are correct, I have the following questions:

The authors note on pg 5 footnote 1 that the modified FISTA algorithm lacks the momentum term. Why, then, are they calling it FISTA? As they explain, FISTA “accelerates the convergence of ISTA by adding a momentum term in each of its iterations.” So this is really a modified ISTA, correct? (I’ll continue to follow their convention of calling it FISTA below, assuming I missed something.)

How does the LISTA regularization term influence performance? The authors are adding two new terms to FISTA inference: first is variance regularization, the second is l2 distance to the predicted LISTA code. This second one seems circular – LISTA is training to emulate the FISTA outputs, but the FISTA outputs are constrained to look like the LISTA outputs. I would like to see more detail on why this was chosen and what happens if they don’t include it (what is the difference in performance on their chosen metrics and the weights learned).

For the reconstruction & denoising experiments, are the images used from the test set? I may have missed this. If not, they should be. I assume they also used the test set for the classification results (fig 9; table 1). Can the authors point me to where they say this in the text?

Is the SDL decoder weight matrix constrained to be the transpose of the encoder weight matrix used for FISTA? From the equations in 2.1 it seems as though the encoding & decoding weights are tied. However in LISTA they are not tied. If they are not tied, it would be useful to investigate how they compare, similar to what was done in [20].

In their words, LISTA was argued to only be designed for “input vectors drawn from the same distribution as our training samples.” Although they do not clarify, it is assumed that they mean the training and test sets are each independently drawn from the same distribution, as is typical in the classification literature. The submitted work has extended LISTA to explicitly test generalization performance in the classification regime with a test set of images. It would be helpful if the authors provided a direct comparison of generalization classification performance between LISTA and iterative inference methods like ISTA or FISTA. That is, without including their variance regularization energy function.

Regarding figure 9, if I am understanding correctly, the top 1 accuracy for 1-shot (1 example per class) is between 35% and 50% for all models. I believe a zero-shot (the network itself is “trained” without labels) K-NN can achieve 95% accuracy [21]. It is my understanding that one-shot classification accuracy for nonlinear encoders is readily also in the high 90s. So am I correct in assuming that the objective here is not to perform well in general in this setting, but simply to compare SDL against VDL? If so, it would be helpful if the authors made this more explicit. It would also be helpful if they extended the curve in fig 9 until the number of training samples is maximal – that is using the full training set. As I mentioned earlier, there is much disagreement in the field as to whether sparse coding is useful for this classification task at all. Extending their results to the domains that are tested by other works will allow the reader to easily compare their experimental setup and model performance to similar alternatives.

**Summary**
It is necessary to investigate each of all of the proposed modifications individually and in an intersectional manner in order to properly assess how they compare. To that end, I would suggest that the authors perform a larger set of ablation experiments.

**References**

VARIANCE NORMALIZATION

[1] P Foldiak (1989) - Adaptive Network for Optimal Linear Feature Extraction

[2] P Foldiak (1990) - Forming Sparse Representations by Local Anti-Hebbian Learning

[3] J Zylberberg, J Murphy, MR DeWeese (2011) - A Sparse Coding Model with Synaptically Local Plasticity and Spiking Neurons Can Account for the Diverse Shapes of V1 Simple Cell Receptive Fields

[4] EVM Dodds, MR DeWeese (2019) - On the Sparse Structure of Natural Sounds and Natural Images- Similarities, Differences, and Implications for Neural Coding

[5] BA Olshausen, DJ Field (1996) - Emergence of Simple-Cell Receptive Field Properties by Learning A Sparse Code For Natural Images

[6] G Barello, AS Charles, JW Pillow (2018) - Sparse-Coding Variational Auto-Encoders

SPARSE CODING IMPROVES CLASSIFICATION

[7] J Mairal, F Bach, J Ponce, G Sapiro, A Zisserman (2008) - Discriminative Learned Dictionaries For Local Image Analysis

[8] J Mairal, F Bach, J Ponce, G Sapiro (2009) - Online Dictionary Learning for Sparse Coding

[9] R Raina, A Battle, H Lee, B Packer, AY Ng (2007) - Self-Taught Learning: Transfer Learning from Unlabeled Data

[10] J Yang, K Yu, Y Gong, T Huang (2009) - Linear Spatial Pyramid Matching Using Sparse Coding for Image Classification

[11] A Coates, H Lee, AY Ng (2011) - An Analysis of Single-Layer Networks in Unsupervised Feature Learning

[12] DM Paiton, CG Frye, SY Lundquist, JD Bowen, R Zarcone, BA Olshausen (2020) - Selectivity and Robustness of Sparse Coding Networks

SPARSE CODING DOES NOT IMPROVE CLASSIFICATION

[13] A Coates, AY Ng (2011) - The Importance of Encoding Versus Training with Sparse Coding and Vector Quantization

[14] R Rigamonti, MA Brown, V Lepetit (2011) - Are Sparse Representations Really Relevant for Image Classification

[15] K Luther, HS Seung (2022) - Sensitivity of Sparse Codes to Image Distortions

LEARNING FROM EQUIVARIANCE

[16] Y Chen, DM Paiton, BA Olshausen (2018) - The Sparse Manifold Transform

[17] P Foldiak (1990) - Learning Invariance from Transformation Sequences

[18] A Hyvarinen, J Hurri, J Vayrynen (2003) - Bubbles- A Unifying Framework for Low-Level Statistical Properties of Natural Image Sequences

[19] L Wiskott, T Sejnowski (2002) - Slow feature analysis-Unsupervised learning of invariances

OTHER REFERENCES

[20] JT Rolfe, Y Lecun (2013) - Discriminative Recurrent Sparse Auto-Encoders

[21] Y LeCun, L Bottou, Y Bengio, P Haffner (1998) - Gradient-Based Learning Applied to Document Recognition

---

> ### Author Response · Authors · 2022-05-20
> **Response to Reviewer G1LD [R2] - Related Literature**
>
> We thank the reviewer for their helpful comments and feedback and for providing an extensive list of references! We are glad that they find the paper generally well-written and they appreciate the extent of our experiments and supplementary information we’ve provided in the appendix. Below we address the questions and issues raised by the reviewer with respect to the related literature and experimental setup.
>
> **Q1. [R2] Related Literature**
>
> **“The idea of using latent variance to regularize sparse codes enforcing a fixed dictionary norm is not new.“**
>
> We find this categorization of our contribution inaccurate. We would like to emphasize that our proposed method applies variance regularization to the latent codes and does not enforce a fixed norm of the dictionary elements.
>
> We would also like to argue in favor of the novelty of our method. In our understanding, works in [1]-[3] are related to ours but aim for decorrelation of the features and maximization of the variance of the features. This strategy differs from what we propose, namely that variance of each latent component is above a certain fixed level.
>
> We thank reviewer R2 for pointing out the strategy of [5], namely “The vector length (gain) of each basis function was adapted over time so as to maintain equal variance on each coefficient”, which essentially normalizes the basis functions. While we find it essentially different from our variance regularization strategy, it is indeed related to our work and we will add it to our related literature section.
>
> **“While somewhat orthogonal, the more modern SVAE model proposed by [6] also includes a variance-based adaptation scheme in the weight update rule and the inference method has constrained latent variance by definition due to the VAE framework.”**
>
> The SVAE (and VAE) frameworks are indeed related to our work. Variance there is used to control the importance of the reconstruction term in the loss (eq 19 in SVAR) as well as to control the variance of the latent codes via the prior distribution (by the nature of the VAE model). We will include them in our related literature section.
>
> We thank the reviewer for the additional references on: research that supports the statement that sparse coding helps with classification [7-12] or refutes it [13-15], as well as research on equivariance [16-19]. We are familiar with several of these works and, as suggested by the reviewer, we are using them and are conducting our own literature review to provide a broader context for our contribution.

---

> > ### Author Response · Authors · 2022-05-20
> > **Response to Reviewer G1LD [R2] - Depth of Experimentation**
> >
> > **Q2. [R2] Depth of experimentation**
> >
> > The reviewer’s description of the SDL setup is not complete and we would like to clarify this. The SDL setup does include a linear decoder (whose columns have a fixed $l_2$ norm) and adopts the standard FISTA inference procedure during training. However, the SDL setup also includes a learned LISTA encoder which is trained to predict the sparse codes obtained through inference with FISTA. Additionally, the term encouraging predictions of the encoder and FISTA to be close to each other (the $\gamma$ term in equation (13)) is also present in the SDL case. After training of the encoder and decoder is completed, the encoder is used to compute sparse codes for given inputs (amortized inference). We employ an encoder in the SDL setup to have a direct and consistent comparison between the SDL and VDL setups. We will clarify all of this in the text.
> >
> > As suggested by the reviewer, the only difference between SDL and SDL-NL is that the decoder in the SDL-NL setup is non-linear.
> >
> > For VDL, VDL-NL, Denoising, and Few-shot classification experiments, the understanding of the reviewer is correct.
> >
> > **Q2.1. [R2] “The authors note on pg 5 footnote 1 that the modified FISTA algorithm lacks the momentum term. Why, then, are they calling it FISTA?”**
> > We thank the reviewer for bringing this up! What we meant to convey in the footnote was that while the momentum terms are still used in the modified version of FISTA, we’re not explicitly introducing more notation for them. We’ll clarify in the text that the gradient step in the modified version of FISTA (eq (9)) is applied to the momentum terms.
> >
> > **Q2.2. [R2] “How does the LISTA regularization term influence performance? Why was it chosen and what happens if it is not included?”**
> >
> > The motivation behind introducing the LISTA regularization term is to bias the codes computed with FISTA towards ones that can be learned easily by the LISTA encoder. Without this term, training of the LISTA encoder becomes harder and a more complex encoder might be needed in order to learn the codes computed by the highly non-linear FISTA algorithm.
> >
> > **Q2.3. [R2] “For the reconstruction & denoising experiments, are the images used from the test set? I assume they also used the test set for the classification results (fig 9; table 1). Can the authors point me to where they say this in the text?”**
> >
> > Yes, we use images from the test set for the reported reconstruction and denoising experiments, as well as for the classification results.  For the reconstruction experiments, we note that “the average PSNR evaluated on the test set over 5 random seeds” on page 7. For the denoising experiments we note “Evaluated on the test set over 5 random seeds.” in the caption of Table 1. We also note “Figure 9 summarizes the classification performance on the test set.“ in the main text and mention ”Test set performance is reported over 5 random seeds” in the Fig 9 caption. We will specify that images in Fig 5 and Fig 6 (examples of denoising) also come from the test set.
> >
> > **Q2.4. [R2] “Is the SDL decoder weight matrix constrained to be the transpose of the encoder weight matrix used for FISTA? From the equations in 2.1 it seems as though the encoding & decoding weights are tied. However in LISTA they are not tied. If they are not tied, it would be useful to investigate how they compare, similar to what was done in [20].”**
> >
> > No, the encoder and decoder’s weights are not tied. To clarify, the equations in section 2.1. do not involve an encoder as the $\mathbf{E}$ notation denotes energy. We will add a visualization of the LISTA encoders bottom layer $U$ similar to that in [20].
> >
> > **Q2.5. [R2] “It would be helpful if the authors provided a direct comparison of generalization classification performance between LISTA and iterative inference methods like ISTA or FISTA.”**
> >
> > We will include the linear classification performance of representations obtained using solely FISTA for inference.
> >
> > **Q2.6. [R2] “So am I correct in assuming that the objective here [Fig 9] is not to perform well in general in this setting, but simply to compare SDL against VDL?”**
> >
> > Yes, the goal of this experiment is to compare pre-trained LISTA encoders from SDL, VDL, SDL-NL, and VDL-NL setups through linear classification. We’ll clarify this in the paper.
> >
> > **Q2.7. [R2] “It would also be helpful if they extended the curve in fig 9 until the number of training samples is maximal – that is using the full training set.”**
> >
> > We will add the classification accuracy when all training samples are used to the paper.

---

> > ### Comment · Reviewer_G1LD · 2022-05-24
> > **Second response**
> >
> > Thank you for the detailed response, and for the time that you have put into this review process. If possible, I would like to see the updated paper with the proposed changes. Below I've added some clarifications.
> >
> > Q1]
> > I appreciate you considering the provided literature. I will qualify my original statement by agreeing in part: I have not seen your exact implementation before -- that is, using the batch-wise coefficient variance as a regularization term in the sparse coding energy function. However, your proposal is a method for achieving homeostasis, which is something that has been explored thoroughly in the neural computation literature. I am reiterating my recommendation that you provide stronger support for your method’s novelty by explicitly discussing how the problem has been solved in the past. The second paragraph of page 2 is not sufficient to convince me that your idea is novel given the similarity to previous works.
> >
> > In an effort to help, let me be more clear about the connection I made while reading your paper: In Olshausen et al 1996, the variance is constrained by adaptively tuning the norm of the filters. This method equalizes the variance of coefficients across neurons using a neuron-specific adaptive learning rule that is informed by the historical variance of prior image presentations. This is, of course, different from what you have proposed, although the connection is clear. I’ll also concede that the insight here is not as readily apparent from reading the paper (the discussion is essentially limited to a sentence in a figure caption), and is much more obvious when one looks at the code for SparseNet itself.
> >
> > Foldiak, 1990, implemented an adaptive thresholding mechanism to achieve homeostasis: delta_t = gamma (y - p), where gamma is a trade-off parameter, y is the neuron activation, and p is "is the specified bit probability", or the desired activation rate. I think you’ll find that this is actually pretty close to what you are doing (although again not exactly the same; he's using a spiking model, etc etc). The threshold of the neuron is analogous to the cutoff for the shrinkage function, which I believe will be shifted by the derivative of your variance tradeoff term.
> >
> > A somewhat more recent discussion on this that might be helpful is the 2008 Neural Computation paper by Laurent Perrinet titled “Role of homeostasis in learning sparse representations.” As you said, none of these works modify the sparse coding energy function as you have – but they’re still relevant for providing context.
> >
> > Q2]
> > Thank you for clarifying the models tested and providing answers to my many other questions. I look forward to seeing SDL models without LISTA included in the results set.

---

> > > ### Author Response · Authors · 2022-06-21
> > > **Follow-up**
> > >
> > > We thank reviewer G1LD [R2] for providing this additional and insightful context on homeostasis and connections to Olshausen et al 1996, Foldiak 1990, and the 2008 Neural Computation paper by Laurent Perrinet! These are all helpful references to place our work in a broader context within the existing literature and we are in the process of incorporating them in the main text.
> > >
> > > We would also like to share that we have provided the additional classification results for SDL FISTA-only models and for models trained on all of the MNIST training data based on our earlier discussion. The details can be found in the "Additional Experiments" post and the supplementary materials.
> > >
> > > We thank the reviewer again for their time and effort to improve our work!

---

### Review · Reviewer_Lkv3 · 2022-05-16

**Summary Of Contributions:**

This paper presents the use of a variance regularization term within the classic sparse coding/dictionary learning formulation. The paper also proposes to integrate a neural encoder to predict the sparse codes produced by the FISTA algorithm directly from the input vectors. The experiments show some promise.


**Broader Impact Concerns:**

I do not see any ethical concerns with this paper. Nevertheless, the paper has provided a paragraph on such concerns already.

**Requested Changes:**

There are minor typos:
a. In (2), there must be a term ||d_i|| = 1 for d_i being the i-th column of D, and \forall i in \set{1,2,...|D|}. The optimization on D is constrained to be in this product space of unit spheres.

b. The FISTA iterations in (6) should be z^(k) instead of x^(k).

c. For the Figure 1, y and D must have arrows going into FISTA algorithm.

Also, please address the above concerns regarding a clear motivation for the need for a new method.

**Strengths And Weaknesses:**

Pros.
A simple idea, the experiments show some minor improvements.

Cons:
As long as the dictionary atoms in D are regularized well (such as using unit norm), I do not see how the l1 norm on the sparse codes will collapse. What is the issue with normalizing the columns of the dictionary that we need a new method to regularize the sparse codes? This is the key contribution of this paper and will need a clear motivation. Regularizing the sparse codes using a statistical measure will lead to a stochastic sparse code, which may not be what one would want for an application.

What is the point of predicting the sparse codes produced by FISTA using a neural network? Perhaps this contribution is besides the main topic of the paper, and could be omitted.

Overall, this paper does not seem to have a mathematically convincing argument for its contributions. Experiments do not bring out the advantages clearly.

---

> ### Author Response · Authors · 2022-05-20
> **Response to Reviewer Lkv3 [R3]**
>
> We thank the reviewer for their comments and helpful suggestions! We are glad that they think the results are promising. Below we summarize and address the questions and issues raised by the reviewer.
>
> **Q1. [R3] “What is the issue with normalizing the columns of the dictionary that we need a new method to regularize the sparse codes?”**
>
> As mentioned in Q1-Q2 [R1], variance regularization applied during training avoids the collapse issue **directly** (by ensuring that the variance across each latent code component for a batch of images is above a fixed threshold) whereas standard $l_2$-normalization of the dictionary elements prevents collapse **indirectly**. Additionally, in the case of a multi-layer decoder and standard $l_2$-normalization of the columns of each layer, the model might be harder to optimize.
>
> **Q2. [R3] “Regularizing the sparse codes using a statistical measure will lead to a stochastic sparse code, which may not be what one would want for an application.”**
>
> Not quite. It is true that our modified version of FISTA relies on batch statistics and is therefore stochastic. However, the term in equation (13) ensures that codes computed during inference with FISTA stay close to the ones predicted by the LISTA encoder. Additionally, after training is complete, codes are computed with the LISTA encoder which is deterministic.
>
> **Q3. [R3] “What is the point of predicting the sparse codes produced by FISTA using a neural network?”**
>
> The motivation for predicting the codes produced by FISTA using a neural network (LISTA encoder) is twofold. Firstly, it removes the need to use batch statistics to infer codes for given inputs. Additionally, the LISTA encoder provides amortized inference, thus reducing the inference cost.
>
> **Q4. [R3] “This paper does not seem to have a mathematically convincing argument for its contributions.”**
>
> Our main motivation is to avoid the collapse problem in sparse coding setups in which the decoder is non-linear. Ensuring that the variance of each latent component is above a fixed threshold is a direct and mathematically-grounded way to achieve this goal. We show empirically that our variance regularization strategy is effective.
>
> **Q5. [R3] “Experiments do not bring out the advantages clearly.”**
>
> We summarize our experimental results as follows. They can be grouped in two categories: qualitative and quantitative.
>
> For the qualitative results, we show that our variance regularization method allows the decoders to learn interpretable and useful features (Gabor-like filters and gratings) similar to those in the standard sparse coding setup, both in the case of a linear and a non-linear dictionary.
>
> For the quantitative results, we show that:
> - In terms of *reconstruction quality*, VDL and VDL-NL models have favorable sparsity to reconstruction quality ratio when compared to SDL and SDL-NL models.
>
> - In terms of *denoising* images corrupted with Gaussian noise, VDL models have comparable performance to SDL models.
>
> - In terms of *linear classification accuracy* of representations learned from the LISTA encoders in the low data regime (with only a few training samples per class available for supervised training), we find that representations from LISTA encoders trained in the VDL-NL setup outperform representations from LISTA encoders trained in SDL, SDL-NL, and VDL (unsupervised) settings, as well as representations from LISTA encoders trained from scratch.
>
> We will make sure to point out more clearly the merits of our method in the paper.
>
> **Q6. [R3] Requested Changes**
>
> a. “In (2), there must be a term ||d_i|| = 1 for d_i being the i-th column of D, and \forall i in \set{1,2,...|D|}.”
>
> Thank you for pointing this out, we’ll include it in the paper.
>
> b. “The FISTA iterations in (6) should be z^(k) instead of x^(k).”
>
> The $x_k$ are the momentum terms used in the FISTA setup and are the correct ones to use for the gradient step.
>
> c. “For the Figure 1, y and D must have arrows going into FISTA algorithm.”
>
> We omitted the arrows from $y$ and $\mathcal{D}$ for a neater diagram. However, we’ll include them if the reviewer believes that it would be beneficial for the readers.

---

> > ### Comment · Reviewer_Lkv3 · 2022-06-10
> > **Second Review**
> >
> > Thank you for responding to the questions I had on the previous draft. I had a fresh read of the paper, and some of the concerns still bother me. Specifically,
> > 1. I do not still see a significant technical contribution that the paper offers. The motivation for proposing to use variance regularization as a way to avoid sparse code collapse appears to be one of many heuristic options that one could come up with, and I do not see how it is better compared to other potential ideas that one could have. More concretely, if the goal is to avoid collapse without regularizing the dictionary atoms, then one could think of adding even simpler penalties such as having the \ell_1 norm of the codes to be greater than some threshold, or having the mean sparse code \mu to have \ell_0 norm of the code dimension (of course, you may need to use some straight-through operator in that case, but in a non-linear decoder setting, that may not be a problem), or any such. Why is the proposed method of variance regularization expected to be better than such simple alternatives?
> >
> > 2. Further, it is unclear to me how the batch-statistic is scalable? For example, if your sparse codes are say thousands of dimensions, how can you scale the batch-size so that the variance across all the sparse codes have some non-zero entry for some dimension of every sparse code? Will having only the variance regularization be sufficient to avoid collapse in such a case? Perhaps a more rigorous proof would be useful to keep such questions away.
> >
> > 3. I also agree with the comments of R1 regarding the use of weight decay. That is a standard method to avoid large values in the learned weights, and a study of the effects of weight decay is inevitable to understand if there is an advantage of variance regularization against standard ways to avoid overfitting of the dictionary.
> >
> > 4. The experiments could be made much stronger by also studying the impact of the dictionary size, the dimensionality of the sparse codes, impact of the batch size, etc.
> >
> > Other comments:
> > 1. The paper writes in the Introduction: "However, it is not trivial to extend this normalization procedure to the case when the decoder D is a multi-layer neural network ...". It is unclear to me what is non-trivial here? Is it the analysis or is it just normalizing the dictionary atoms? Why should the latter be non-trivial? And if it is the former, then how is the contribution fixing it?
> >
> > 2. I would still think Figure 1 should use y and D as inputs to FISTA, perhaps you could use dotted-lines to show this to avoid clutter. You could also split the figure into two figures, one for training of the encoders, and other for the sparse codes.

---

> > > ### Author Response · Authors · 2022-06-21
> > > **Response to Second Review**
> > >
> > > We thank reviewer Lkv3 [R3] for the updated review and additional questions! We address them below.
> > >
> > > **[1 & 3] Why is the proposed method of variance regularization expected to be better than such simple alternatives [$l_1$,  $l_0$ strategies]? I also agree with the comments of R1 regarding the use of weight decay.**
> > >
> > > We consider our proposed variance regularization strategy (assuring that the variance across every latent dimension is above a fixed threshold) a simple and intuitive way to prevent collapse in the codes. It is differentiable (as opposed to the $l_0$ strategy) and it also has the added benefit of encouraging all latent dimensions to be used which is not guaranteed by the proposed alternatives. That being said, the variance regularization strategy does not come at the cost of codes with more active components - as evident from Fig 4 in the main paper and Fig 3 in the supplementary material, models trained using variance regularization produce sparser codes which generate higher-quality reconstructions. Early on, we considered different regularization strategies, for example, ensuring that each individual code has variance greater than a given threshold. However, this strategy wasn’t able to produce interpretable features in the decoder as the ones seen in standard dictionary learning.
> > >
> > > We include additional experimental results on models trained with weight decay in the supplementary material. In particular, in terms of the trade-off between sparsity (measured by average percentage of inactive (zero) code components) and reconstruction quality (measured by PSNR), codes from models with a non-linear decoder trained using variance regularization perform better than codes from all other models (including those utilizing weight decay). Also, in the case of sparse autoencoders with a linear dictionary trained with weight decay regularization on the MNIST dataset, we observe that the higher the sparsity regularization, the more of the dictionary atoms die off, i.e. they do not contain interpretable features. This is not the case in models trained with our variance regularization strategy (as seen in the Appendix in Fig 11c,d).
> > >
> > > **[2]. How is the batch-statistic is scalable? E.g., if your sparse codes are thousands of dimensions, how can you scale the batch-size so that the variance across all the sparse codes have some non-zero entry for some dimension of every sparse code? Will having only the variance regularization be sufficient to avoid collapse in such a case? Perhaps a more rigorous proof would be useful to keep such questions away.**
> > >
> > > We expect that in the case of very large-dimensional codes, variance regularization would still be effective to prevent collapse. Even if some latent dimensions are not being used for a given batch of images, variance regularization should ensure that the ones that do get used are maintaining a desired level of variance. Assuming that the dataset is large (and versatile) enough, we expect that the majority of the latent dimensions will be used eventually. Additionally, initializing codes in FISTA with the predictions of the encoder should further help with utilizing all latent dimensions, especially early in training, when the encoder’s predictions are random.
> > >
> > > **4. The experiments could be made much stronger by also studying the impact of the dictionary size, the dimensionality of the sparse codes, impact of the batch size, etc.**
> > >
> > > We consider this point is related to question 2 above about the scalability of the batch size when the codes have a large dimension. We agree that adding a set of experiments which analyze the effect of the batch size on models trained with variance regularization would be beneficial and may include such a set of experiments if time permits it. However, we believe that even without it, one of our main claims, namely that variance regularization is a successful strategy for training multi-layer sparse autoencoders, is still well supported with the existing set of experiments.
> > >
> > > **Add. 1. "However, it is not trivial to extend this normalization procedure to the case when the decoder D is a multi-layer neural network ...". It is unclear to me what is non-trivial here?**
> > >
> > > What we mean is that it is not clear what normalization strategy for non-linear decoders would result in sparse representations which are useful in terms of downstream applications - we will clarify this in the text. In our work, we show that the variance regularization strategy + LISTA encoder produce sparse representations which are advantageous in terms of the trade off between sparsity level and reconstruction quality as well as the classification accuracy with few labeled training samples.
> > >
> > > **Add. 2. I would still think Fig 1 should use y and D as inputs to FISTA, perhaps you could use dotted-lines to show this to avoid clutter.**
> > >
> > > Thank you! We believe dotted lines will work well in the revised version.
> > >
> > > We thank the reviewer again for their time and effort!

---

### Author Response · Authors · 2022-05-20
**Shared response to reviewers**

We thank the reviewers for their helpful comments and valuable feedback! We are glad that the reviewers find the idea of regularizing the sparse codes novel [**unXH**], find the paper generally well-written [**G1LD**], appreciate the extent of our experiments and supplementary information we have provided in the appendix [**G1LD**], and state that the experiments “show some promise” [**Lkv3**].

We have summarized and addressed the questions raised by the three reviewers in relation to the motivation for our work, the related literature, the experimental setup and results, and provide a response to each reviewer separately.

For convenience, we adopt the following common notation in our responses:

Reviewer **unXH** -> **R1**

Reviewer **G1LD** -> **R2**

Reviewer **Lkv3** -> **R3**

**VDL** - sparse autoencoder in which variance regularization is applied to the codes during inference with FISTA and the decoder is linear.

**VDL-NL** - sparse autoencoder in which variance regularization is applied to the codes during inference with FISTA and the decoder is a fully connected network with one hidden layer and ReLU non-linearity.

**SDL** - sparse autoencoder in which there is no variance regularization during inference with FISTA, the decoder is linear and its columns are restricted to have unit norm.

**SDL-NL** - sparse autoencoder in which there is no variance regularization during inference with FISTA, the decoder is a fully connected network with one hidden layer and ReLU non-linearity and the columns of each layer in the decoder have unit norm.

All of the models above have a **LISTA**-inspired encoder which is used for amortized inference after training is finished.

Again, we are very grateful to the reviewers for their input on improving our work!

---

### Author Response · Authors · 2022-06-21
**Additional Experiments**

We sincerely thank the reviewers for their helpful feedback on ways to improve our work!

As requested by the reviewers, we are sharing results on additional experiments which compare different strategies for preventing collapse. In particular, we compare autoencoders trained with weight decay regularization applied to the decoder to autoencoders trained using our proposed variance regularization strategy and to autoencoders trained using standard normalization of the decoder’s weights. Additionally, we train decoder-only models which use the standard FISTA algorithm for inference and use standard normalization of the decoder’s weights. We explain the experimental results below but would first like to summarize the notation for the different models we use:

**VDL** - LISTA encoder, linear decoder, no normalization of the decoder’s weights, variance regularization in FISTA during training

**VD-NL** - LISTA encoder, decoder: fully connected network with one hidden layer, no normalization of the decoder’s weights, variance regularization in FISTA during training

**SDL** - LISTA encoder, linear decoder, the decoder’s columns are restricted to have unit norm, no variance regularization in FISTA

**SDL-NL** - LISTA encoder, decoder: fully connected network with one hidden layer, the columns of each layer in the decoder have unit norm, no variance regularization in FISTA

**FO** - no encoder, linear decoder, the decoder’s columns are restricted to have unit norm, no variance regularization in FISTA

**FO-NL** - no encoder, decoder: fully connected network with one hidden layer, the columns of each layer in the decoder have unit norm, no variance regularization in FISTA

**WD** - LISTA encoder, linear decoder, weight decay applied to the decoder’s weights, no variance regularization in FISTA

**WD-NL** - LISTA encoder, decoder: fully connected network with one hidden layer, weight decay applied to the decoder’s weights, no variance regularization in FISTA

In models a LISTA encoder:
- during training, there is a regularization term in FISTA encouraging codes to be close to the encoder’s predictions,
- after training is finished, the LISTA encoder is used for amortized inference after training is finished.

In models without an encoder, regular FISTA is used to compute the codes during training and after training is finished.


**Classification Accuracy for FISTA-only models [R2 - G1LD]**

As discussed, we:
 - include the test classification accuracy on MNIST for FISTA-only models (FO, FO-NL) trained with different numbers of training samples per class;
- plot the test classification accuracy when all training samples are used for all the models.

The updated figures for top-1 and top-3 test set accuracies are Fig 1 and Fig 2 in the supplementary material. Both plots show that the VDL-NL model still has an advantage over the other models when the number of training samples is 1, 2, 5, and 10.

**Models trained with Weight Decay [R1 - unXH, R3 - Lkv3]**

We ran WD and WD-NL experiments as a baseline in which weight decay is used to prevent collapse (instead of variance regularization or normalization of the decoder’s weights).

We update the regret curve from Fig 4 in the paper capturing the trade-off between sparsity (measured by average percentage of inactive (zero) code components) and reconstruction quality (measured by PSNR) for codes computed from VDL, VDL-NL, SDL, SDL-NL models by adding results for codes from FO, FO-NL, WD, and WD-NL models. All models are trained on the MNIST dataset with varying levels of sparsity regularization. We still observe that for higher levels of sparsity, codes from the VDL-NL models produce better reconstructions than codes from the other models.

Additionally, we notice that for WD models trained on MNIST data, the higher the sparsity regularization, the more of the dictionary atoms die off, i.e. they do not contain interpretable features. This is not the case in models trained with our variance regularization strategy (as seen in the Appendix in Fig 11c and 11d).

We believe these additional experiments strengthen our main argument that variance regularization is a successful strategy for training sparse autoencoders. We would like to thank the reviewers again for their valuable feedback!

---

### Decision · Action_Editors · 2022-06-29

**Recommendation:** Accept with minor revision

**Comment:**

The submission introduces a new sparse coding approach which avoids collapse in the codes without having to regularize the decoder. Instead, variance regularization is applied to each latent code component in the form of a squared hinge term encouraging the code component's variance over a given mini-batch to reach some pre-determined threshold value.  Since inference now introduces a dependency across latent codes, the submission also proposes to train an encoder which maps inputs to their corresponding latent codes.

The proposed approach (VDL) is compared against the usual sparse coding approach (SDL) which enforces a constant L2 norm on the decoder weight columns. Various sparsity regularization coefficients are used during training, and the approaches are evaluated in terms of peak signal-to-noise ratio (PSNR) for MNIST and ImageNet patches, and VDL is found to retain better PSNR at larger code sparsity values. The models are also evaluated on two downstream tasks, namely image denoising and MNIST classification in the low data regime. For the latter, performance is measured in terms of the linear separability of the latent codes, and VDL with a non-linear decoder is shown to perform better than the alternatives in the very low data regime.

Reviewers enjoy the writing quality (unXH, G1LD) and clarity (G1LD), and note that the proposed approach is straightforward (Lkv3) and the evaluation ventures beyond reconstruction and denoising to also consider classification in the low-data regime (G1LD). The main concerns they raise are:

- The submission lacks references to multiple related works (G1LD). Reviewer G1LD provides a comprehensive list of citations which the authors acknowledge and promise to incorporate in the submission. Reviewer G1LD concedes that they haven't seen the specific variance regularization implementation proposed in the submission before. This point has been addressed to their satisfaction, as their official recommendation leans towards acceptance (provided the promised changes are materialized in a submission update).
- It's unclear why one would prefer the proposed approach over alternatives such as weight decay or weight normalization (unXH, Lkv3). The authors clarified that the proposed approach is a direct way of preventing latent code collapse (as opposed to the indirect alternatives), and that extending L2-normalization to the non-linear decoder case might be harder to optimize. The authors also ran additional experiments comparing VDL with weight decay which shows that VDL outperforms it in terms of PSNR on MNIST, and reported observing more dictionary atoms dying off at higher sparsity levels when using weight decay. Reviewer unXH remains unconvinced of the advantages of VDL over simple alternatives. Reviewer G1LD points out that although the performance improvements observed are modest, the submission's value to the research community resides in its extensive testing across the different model variations with consistent methods for parameter selection, architecture, etc. I agree with Reviewer G1LD, given that TMLR "emphasizes technical correctness over subjective significance"; to me the proposed approach is an interesting alternative for avoiding latent code collapse, and the empirical evaluation demonstrates that it performs as advertised.

Given the above, I recommend accepting the submission with minor revision. The revision should incorporate the expanded discussion on related work requested by Reviewer G1LD and the additional experiments currently contained in the supplementary material.

---

> ### Author Response · Authors · 2022-07-05
> **Re: Decision by Paper51 Action Editors**
>
> We thank the Action Editors for their recommendation of acceptance with minor revision and thank the Reviewers again for their helpful feedback! We plan to include the requested changes in the final version of the paper.